# Towards scientific discovery with dictionary learning: Extracting biological concepts from microscopy foundation models

## Abstract

Dictionary learning (DL) has emerged as a powerful interpretability tool for large language models. By extracting known concepts (e.g., Golden-Gate Bridge) from human-interpretable data (e.g., text), sparse DL can elucidate a model's inner workings. In this work, we ask if DL can also be used to discover *unknown* concepts from less human-interpretable scientific data (e.g., cell images), ultimately enabling modern approaches to scientific discovery. As a first step, we use DL algorithms to study microscopy foundation models trained on multi-cell image data, where little prior knowledge exists regarding which high-level concepts should arise. We show that sparse dictionaries indeed extract biologically-meaningful concepts such as cell type and genetic perturbation type. We also propose a new DL algorithm, Iterative Codebook Feature Learning (ICFL) and combine it with a pre-processing step which uses PCA whitening from a control dataset. In our experiments, we demonstrate that both ICFL and PCA improve the selectivity or "monosemanticity" of extracted features compared to TopK sparse autoencoders.

## 1 Introduction

Large scale machine learning systems are extremely effective at generating realistic text and images. However, these models remain black boxes: it is difficult to understand how they produce such detailed reconstructions, and to what extent they encode semantic information about the target domain in their internal representations. One approach to better understanding these models is to investigate how models encode and use high-level, human-interpretable concepts. A challenge to this endeavor is the "superposition hypothesis" (Bricken et al. 2023), which states that neural networks encode many more concepts than they have neurons, and as a result, one cannot understand the model by inspecting individual neuron. One hypothesis for how neurons encode multiple concepts at once is that they are low-dimensional projections of some high-dimensional, sparse feature space. Quite surprisingly, there is now a large body of empirical evidence that supports this hypothesis in language models (Mikolov et al., 2013; Elhage et al., 2022; Park et al., 2023), games (Nanda et al., 2023) and multimodal vision models (Rao et al., 2024), by showing that high-level features are typically predictable via *linear* probing. Further, recent work has shown that model representations can be decomposed into human-interpretable concepts using a dictionary learning model, estimated via sparse autoencoders (Templeton, 2024; Rajamanoharan et al., 2024b;a; Gao et al., 2024).

However, all of these successes have relied on some form of text supervision, either directly through next-token prediction or indirectly via contrastive objectives like CLIP (Radford et al., 2021), which align text and image representations. Further, these successes appear in domains which are naturally human-interpretable (i.e. text, games and natural images), and as a result, one may worry that high-level features can be extracted only in settings that we already understand. This raises a natural question: can we extract similarly meaningful high-level concepts from completely unsupervised models in domains where we lack strong prior knowledge? For example, in computational biology, masked autoencoders (MAE) trained on cellular microscopy images have been shown to be very effective at learning representations that recover known biological relationships (Kraus et al., 2024). However, it is not known whether analogous high-level concepts can be extracted from these large MAEs. These settings are precisely where extracting high-level concepts could be most valuable: given that models can detect subtle differences in images (even those that are very challenging for

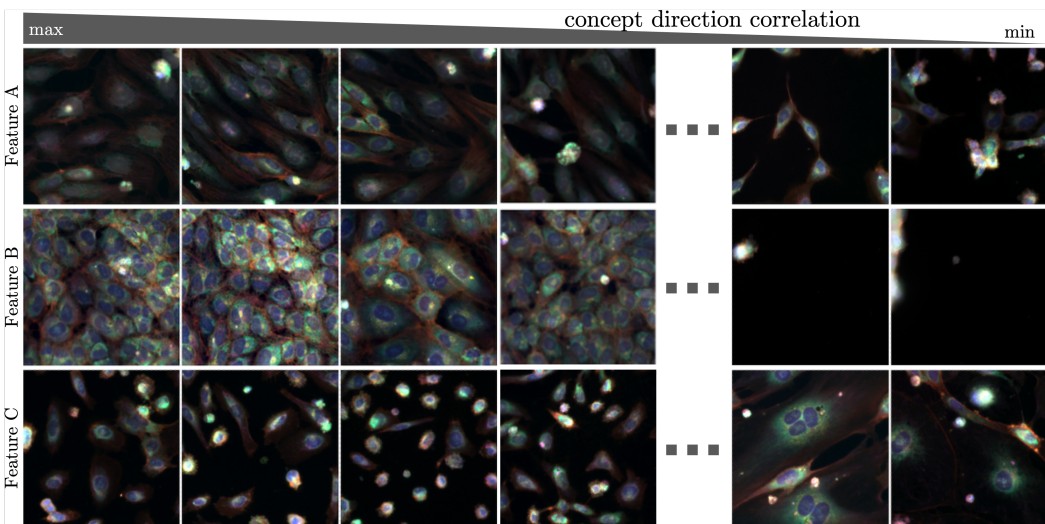

Figure 1: Cell images ranked according to the correlation strength with three selected features learned by our dictionary learning algorithm. Each feature captures distinct cellular morphologies: Feature $A$ activates for cells with an elongated, spindle-like shape (left) and anti-correlates for sparser or aggregated cells (right); Feature $B$ activates for cells that are densely packed with closely arranged nuclei (left) and deactivates when cell density drops (right); and Feature $C$ activates for small-shaped, compact, brights cells without cell-cell contacts almost entirely made up from just nuclei (left), in contrast to multi-nucleated cells which occupy larger areas (right).

human experts to interpret), we might hope that we can use these techniques to better understand subtle differences.

We study the extraction of high-level concepts from large-scale MAEs trained on microscopy images of cells that have been perturbed in genetic and small molecule perturbations screens (Fay et al., 2023). Understanding the morphological changes induced by genetic and small molecule perturbations is an inherently difficult and fundamental problem that plays a crucial role in drug discovery (Celik et al., 2022). Recent progress in this field using machine learning has been made by building similarity maps of genetic perturbations via cosine-similarities of post-processed representations from MAEs (Kraus et al., 2024; Celik et al., 2022; Lazar et al., 2024). However, a limitation of these deep learning-based methods is that we only gain limited insights about the morphological changes arising from the perturbations: we can tell whether two perturbations are similar (or dissimilar) via cosine similarity, but we cannot tell *why* (or the ways in which) they are different. That is, we collapse the multidimensional similarities and dissimilarities down to a single score.

In this paper, we train dictionary learners on top of intermediate representations of large-scale MAEs (Kraus et al., 2024) and find features correlated with single concepts such as individual cell types or genetic perturbations in an unsupervised manner. Moreover, via linear probing, we show that the reconstructed representations from the sparse features preserve significant amounts of biologically-meaningful information. Through this research, we make several key contributions:

- We show that dictionary learning can be used to extract biologically-meaningful concepts from microscopy foundation models (see Figure 1), opening the path to scientific discovery using tools from mechanistic interpretability.

- We propose a new dictionary learning algorithm—Iterative Codebook Feature Learning (ICFL)— which naturally avoids "dead" features (Section 4).

- We further show how PCA whitening on a control dataset can act as a form of weak supervision for dictionary learning (Section 5), resulting in more meaningful features.

- We demonstrate empirically that both ICFL and PCA improve the selectivity or "monosemanticity" of extracted features compared to TopK sparse autoencoders (Section 6).

---

**Algorithm 1** Iterative Codebook Feature Learning

---

1: **Input:** Parameters $W_{\text{dec}}, b_{\text{pre}}$; model representation $x$; # sparse features $K$ and iterations $J$
2: Initialize $x^{(1)} := x - b_{\text{pre}}$
3: **for** $t = 1$ **to** $J$ **do**
4:     Select top $K$ columns of $W_{\text{dec}}$ which maximize $\langle W_{\text{dec},m}, x^{(t)} \rangle$
5:     Solve $z^{(t)} = \arg\min_z \|x^{(t)} - W_{\text{dec}} z\|_2^2$ with $z$ non-zero only for selected columns
6:     Update $x^{(t+1)} := x^{(t)} - W_{\text{dec}} z^{(t)}$
7: **end for**
8: **Output:** Sparse features $z := \sum_{t=1}^{J} z^{(t)}$

---

## 2 RELATED WORK

The disentanglement and causal representation literature (CRL) share the goal of learning high-level, interpretable concepts (Bengio et al., 2013; Kulkarni et al., 2015; Higgins et al., 2017; Chen et al., 2016; Eastwood & Williams, 2018; Schölkopf et al., 2021). Two key differences with the dictionary learning approach are: (i) disentanglement/CRL methods consider low-dimensional representations to capture the factors of variation in data, whereas overcomplete dictionary learning seeks a higher-dimensional representation to capture a large set of sparsely-firing concepts; and (ii) disentanglement/CRL methods aim to be inherently interpretable, whereas this paper considers a post-hoc approach to interpret pre-trained models. Related work on post-hoc explainability also learns "concept vectors" in neural network internal states (Kim et al., 2018; Ghorbani et al., 2019); a key difference is that these methods use class-labeled data, whereas this paper uses an unsupervised approach to discover concepts. Additionally, feature-visualization works aim to interpret internal states/neurons by finding the data points (or gradient-optimized inputs) that lead to maximal activation (Mordvintsev et al., 2015; Olah et al., 2017; Borowski et al., 2021).

## 3 BACKGROUND

**The superposition hypothesis.** Let $x_i \in \mathbb{R}^d$ denote a representation for token $i$; as an example, $x_i$ may be the embedding of token $i$ after a transformer layer. Bricken et al. (2023) hypothesize that (i) such token representations $x_i \in \mathbb{R}^d$ are linear combinations of concepts; (ii) the number of available concepts $M$ significantly exceed the dimension of the representation $d$; and (iii) each token representation is the sum of a sparse set of concepts. These desiderata are satisfied by the following model that is widely studied in compressed sensing and dictionary learning:

$$x_i \approx W z_i \qquad \text{where } \|z_i\|_0 \ll d \tag{1}$$

where $W \in \mathbb{R}^{d \times M}$ is a latent concept matrix and $z_i \in \mathbb{R}^M$ is a sparse latent concept-selector (resp. feature) vector.

**Feature learning using TopK SAEs.** Given a set of token representations $\{x_i\}_{i=1}^N$, learning both $W$ and $\{z_i\}_{i=1}^N$ is a *dictionary learning* or *sparse coding* problem Olshausen & Field (1997), with a long history of works proposing efficient algorithms with provable guarantees (Aharon et al., 2006; Arora et al., 2014; 2015). In the context of mechanistic interpretability, the dominant choice for learning these parameters are two-layer sparse autoencoders. In this paper, we compare to the state-of-the-art method called TopK SAE, originally proposed by Makhzani & Frey (2013) and recently studied by Gao et al. (2024). Following their notation, the model is:

$$x_i = W_{\text{dec}} z_i + b_{\text{pre}}, \quad \text{with } z_i = \text{TopK}(W_{\text{enc}} x_i - b_{\text{pre}})$$

where $\text{TopK}(\cdot)$ is an operator that sets all but the $K$ largest elements to zero. The parameters $\{W_{\text{dec}}, W_{\text{enc}}, b_{\text{pre}}\}$ are learned by minimizing the reconstruction loss:

$$L(W, b) := \sum_i \|x_i - \widehat{x}_i\|_2^2, \quad \text{where } \widehat{x}_i = W_{\text{dec}} \text{TopK}(W_{\text{enc}} x_i - b_{\text{pre}}) + b_{\text{pre}} \tag{2}$$

A problem with the above optimization is that some concept vectors $W_{\text{dec},m}$ are barely used; that is, features $z_{im} = 0$ for almost all $i \in [N]$. This is called the "dead feature" phenomenon. To reduce the amount of dead features, Gao et al. (2024) introduce an additional reconstruction error term using only these concept vectors to encourage their usage in the model (see Table 1).

## 4 ITERATIVE CODEBOOK FEATURE LEARNING (ICFL)

Sparse autoencoders such as TopK SAEs face two major limitations: (i) they require regularization to avoid "dead features" after training (Gao et al., 2024; Bricken et al., 2023) and (ii) some concepts may be overrepresented in the samples $\{x_i\}_{i=1}^N$, biasing the estimation. To overcome these limitations, we propose Iterative Codebook Feature Learning (ICFL). ICFL retains the decoder of TopK SAEs, however, instead of using an encoder to learn the features $z$, ICFL updates $z$ using a variant of the orthogonal matching pursuit algorithm of Mallat & Zhang (1993) as described in Algorithm 1. Specifically, given the current decoder/feature matrix $W_{\text{dec}}$, we first select the top-$k$ columns most aligned with $x^{(1)} = x$. Then, we learn the features $z^{(1)}$ that best reconstruct $x \approx W_{\text{dec}} z^{(1)}$, using only these columns (i.e. $z^{(1)}$ is $K$-sparse). Next, to obtain $z^{(2)}$, we repeat this step, but replace $x$ with the residual $x^{(2)} = x - W_{\text{dec}} z^{(1)}$. Repeating this process, the final output $z$ is taken to be $z = \sum_{t=1}^J z^{(t)}$. Consequently, $z$ is at most $Jk$-sparse.

The key idea of ICFL is that early iterations subtract dominant concepts from $x$, allowing the algorithm in later iterations to select a broader set of concepts that are not as correlated with the main concepts in $x$. After updating $z$ as detailed in Algorithm 1, the decoder parameters $\{W_{\text{dec}}, b_{\text{pre}}\}$ are updated to minimize the reconstruction loss from equation 2 with $\widehat{x} = W_{\text{dec}} z + b_{\text{pre}}$. As $z$ is fixed in this gradient step, the algorithm does not propagate gradients through $z$. Consequently, the algorithm results in very few "dead" features. As a result, we do not require any additional regularization to address this "dead feature" issue that often hinders SAEs, as shown in Table 1.

In practice, we leverage random resets to ensure that the columns of $W_{\text{dec}}$ are not too correlated. To prevent the collapse of the feature directions (columns of $W_{\text{dec}}$), after every 100 stochastic gradient descent steps, we take every pair of columns of $W_{\text{dec}}$ that have cosine-similarity above 0.9 and randomly initialize one of the pairs with a vector selected uniformly at random from the hypersphere. Before running Algorithm 1, we always center the representations by the average representation with unperturbed samples from the control distribution. By doing so, we center the representations such that the origin represents the unperturbed state. Finally, we normalize the representations before applying the dictionary learner.

|      | w/o  | w/   |
|------|------|------|
| ICFL | 55   | 341  |
| TopK | 7640 | 8026 |

Table 1: The number of "dead features" (out of 8192) that have been activated less than a fraction of $10^{-5}$ many times during the last 1000 training steps, for both TopK and ICFL with and without PCA whitening (see Section 5).

## 5 EXPERIMENTAL SETUP

**Data source and foundation model**   We evaluated our dictionary learning approach on two large-scale masked autoencoders trained on cellular microscopy Cell Painting image data using 256x256x6 pixel crops as input and a patch size of 8, following the same procedures as those described in Kraus et al. (2024). These models were trained on data from multiple cell types that were perturbed with both CRISPR gene-knockouts and small molecule perturbations. Both models used the architecture hyperparameters from Kraus et al. (2024), with the smaller of the two using the ViT-L/8 configuration, while the larger model used the ViT-G/8 configuration. We refer to these models as *MAE-L* and *MAE-G*, respectively. We obtain a single token per input crop by aggregating all patch tokens (excluding the class token). For both the residual stream and the attention output (after the out-projection), the dimension $d$ of the tokens (representations) are 1024 and 1664 for MAE-L and MAE-G, respectively. All the visualizations used Cell Painting microscopy images from the public RxRx1 (Sypetkowski et al., 2023) and RxRx3 (Fay et al., 2023) datasets.

We extract the tokens from layer 16 (*MAE-L*) and layer 33 (*MAE-G*), respectively. The motivation for using intermediate instead of final layers is that these tokens are more-likely to capture abstract high level concepts that are *internally used* by the model to solve the SSL task (Alkin et al., 2024). We selected this layer by finding the layer which maximized linear probing performance on the functional group tasked (described below) from the original embeddings.

**Preserving linear probing signals**   To investigate whether the features found by sparse dictionary learning retain important information from the original representation, we define five different classification tasks, summarized in Table 2. For each classification task, we use a separate (potentially overlapping) dataset and split it into train and test data to distinguish labels across:

| Task | Cell Type | Experiment Batch | siRNA Perturbation | CRISPR Perturbation | Functional Gene Group |
|---|---|---|---|---|---|
| # Classes | 23 | 272 | 1 138 | 5 | 39 |
| # Samples | 110,971 | 80,000 | 81,224 | 79,555 | 57,863 |
| Bal. Test Acc. | 97.2% | 87.8% | 51.6% | 94.6% | 32.1% |

Table 2: The five classification tasks and the test bal. acc. for linear probes trained on well-level aggregated representations from the residual stream from an intermediate layer from *MAE-G*.

(1) 23 different cell types which are almost perfectly distinguishable via linear classification.

(2) 272 different experiment batches. Even in controlled conditions, subtle changes in experimental conditions can induce strong *batch effects*, *i.e.* changes in experimental outcomes due to experiment-specific variations unrelated to the perturbation that is being tested.

(3) 1138 siRNA perturbations from the RxRx1 dataset (Sypetkowski et al., 2023), where the single-gene expression (i.e. gene mRNA level) is partially (or completely) silenced using short interfering (si-)RNA. siRNA targets the gene mRNA for destruction via the RNA interference pathway (Tuschl, 2001). As the extent of siRNA knock-downs is hard to quantify and prone to significant but consistent off-target effects, we also evaluated:

(4) 5 single-gene CRISPR perturbation knockouts which induce strong and consistent morphological profiles across cell types, known as "perturbation signal benchmarks" (Celik et al., 2024). Unlike the siRNA approach, CRISPR cuts the gene DNA directly, which induces mutation in the sequence and represses the gene function. To evaluate whether our method retrieves signal which corresponds to similar phenotypes, we also assessed:

(5) 39 functional gene groups composed of CRISPR single-gene knockouts categorized by phenotypic relationships between the genes, including major protein complexes, metabolic and signaling pathways. Each gene group targets similar or related cellular process, which results in inducing morphologically similar changes in the cells (Celik et al., 2022).

To remove the impact of spurious correlations between perturbations and batch effects on the test accuracy, we always use mutually exclusive experiments for test and train data, except for (ii) where the task is to predict the experiment. Except for (i), all classification tasks use HUVEC cells and always use well-level aggregated representations: that is, we take the mean over tokens from all 36 non-edge crops from an image of a given well of cells. Because some of the classes are heavily imbalanced (particularly for Task (1)), we always report the *balanced test accuracy* and train our linear probes using logistic regression on a class-balanced cross-entropy loss.

**PCA whitening using a control dataset** As dictionary learners seek to minimize the Euclidean distance between the model representations $x$ and their reconstructions $\hat{x} = Wz$, the learned features $z$ are naturally biased towards capturing the dominant directions in the data (i.e., those that explain the most variance). Unfortunately, these directions often do not align with meaningful concepts. To address this, we use a dataset of control samples as a form of weak supervision, downweighting dominant directions in this control dataset as we know they do not correspond to the biological perturbations of interest. In particular, we learn a PCA-and-centerscale transform on this control dataset and apply it to the entire dataset *before normalization*. For our multi-cell data, unperturbed HUVEC-cell images act as our control dataset. Note that similar PCA whitening on a control dataset has been used to improve the quality of the learned multi-cell image representations (Kraus et al., 2024).

**Training the DL models** By default, we always choose a sparsity of $K = 100$ for TopK SAEs and $J = 20, k = 5$ (resulting in a max sparsity of 100) for ICFL as described in Section 4, and use a total of 8192 features. Unless otherwise specified, we always apply the PCA whitening described in Section 5 and use representations from the residual stream. We train the sparse autoencoders using 40M tokens (one token per crop) with a batch size of 8192 for 300k iterations. Our learning rate is $5 \times 10^{-5}$ for all experiments. Similar to Gao et al. (2024), we observed that changing the learning rate has a limited impact on the outcome. We present an ablation for the learning rate in Appendix C.

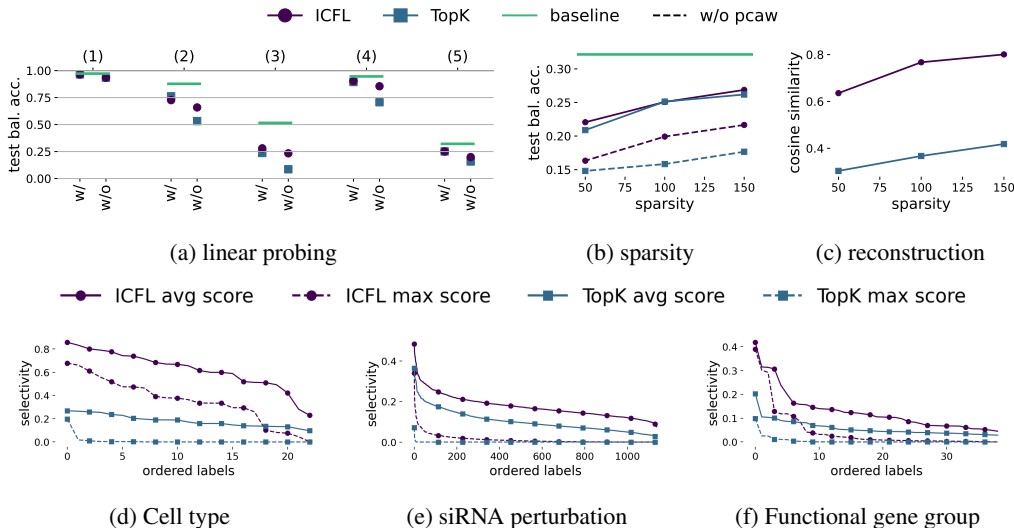

Figure 2: **Top row:** a) Test bal. acc. of linear probes trained on the original representation (solid line) and reconstructions from ICFL and TopK SAEs in combination with PCA whitening and with out. b) Test bal. acc. as a function of the sparsity (dashed line is the original representation) for classification Task 5. c) Cosine similarity of reconstruction and original representations as a function of sparsity for tokens from a hold-out validation dataset. **Bottom row:** The highest selectivity scores among all features for each label. We separately order the labels for each line starting with the maximum score. We plot the avg (solid) and max (dashed) selectivity scores.

# 6 EXPERIMENTAL RESULTS

In this section we present our experimental results. If not further specified, we always use features extracted from ICFL in combination with PCA whitening.

## 6.1 DICTIONARY FEATURES ARE CORRELATED WITH BIOLOGICAL CONCEPTS

**Preserving linear probing signals** By comparing linear probes on the representations and reconstructions from ICFL sparse features, we can measure how much "biologically-relevant" information is lost when extracting sparse features. Figure 2a shows that almost the entire signal is preserved for simple concepts such as cell types (1), batch effects (2) and perturbations with strong morphological changes (4). For the difficult tasks of distinguishing between many genetic perturbations (3,5), a substantial amount of the linear signal is preserved. Both TopK SAEs and ICFL features yield a similar linear probing accuracy, while we can see a clear drop if no PCA whitening is used during pre-processing. We further present in Figure 2b an ablation for the sparsity of the extracted feature vector. While increasing the number of non-zeros improves the accuracy, the effect is limited compared to PCA whitening.

**Reconstruction loss** To evaluate the quality of unsupervised DL, the cosine similarity (or $\ell_2$-error) has been often used as a benchmark (Rajamanoharan et al., 2024a; Gao et al., 2024). Figure 2c shows that the reconstruction quality of ICFL is much higher than TopK SAE for the same sparsity constraints when using PCA whitening. We provide further ablations in Appendix C.

**Selectivity of features for biological concepts** As a third experiment, we investigate how strongly correlated the features are with labels from the classification tasks in Table 2. For each dataset associated with a classification task, we extract from every image a feature vector using the center crop as input to the MAE. For each feature, we then compute two selectivity scores: the **avg selectivity** score, which is the % of times that the feature is active given that label $i$ occurs minus the % of times the feature is active given any other label. As a stronger notion of correlation, we also use the **max selectivity** score, that subtracts the maximum % for any other label. The selectivity score has been originally proposed in the context of neuroscience (Hubel & Wiesel, 1968) and has also been used by Madan et al. (2022) to measure the "monosemanticity" of neurons.

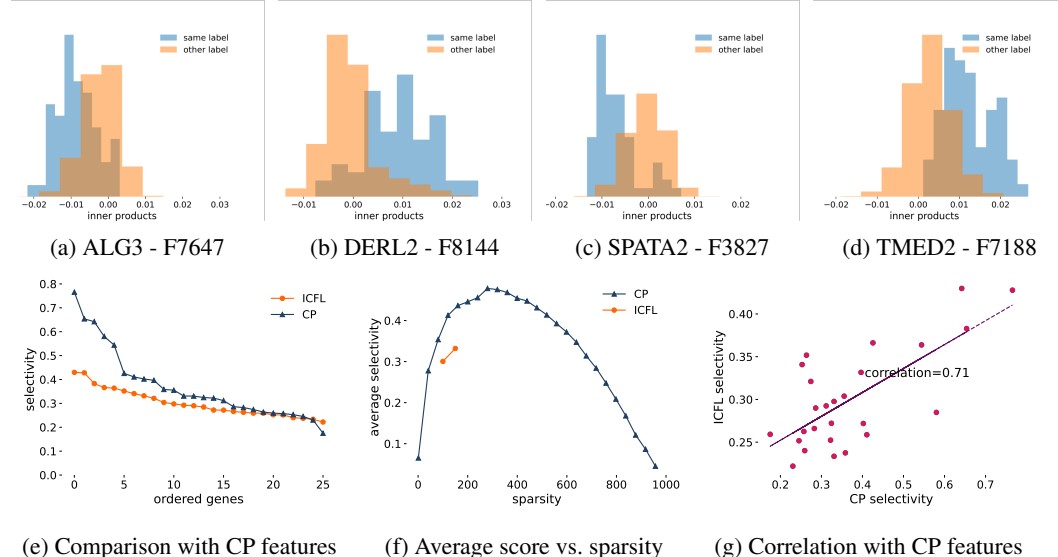

(a) ALG3 - F7647    (b) DERL2 - F8144    (c) SPATA2 - F3827    (d) TMED2 - F7188

(e) Comparison with CP features    (f) Average score vs. sparsity    (g) Correlation with CP features

Figure 3: **Top row:** Cosine-similarity histograms for selected pairs of representations from perturbations from Task 3 and features directions (as shown in the caption), given that its associated perturbation is applied (blue) and that any other perturbation is applied (orange). **Bottom row:** Comparison of the average selectivity score of features from CP and ICFL. a) Maximum average selectivity scores for each label, displayed in descending order. b) The scores from (a) averaged across labels at different thresholds for CP and sparsity levels for ICFL, as a function of the average number of non-zero values. c) Correlation of maximum average selectivity scores for each label between CP and ICFL.

We plot in Figure 2d-2f the selectivity scores for both ICFL features and TopK SAEs. We see that ICFL features consistently achieve higher selectivity scores than TopK SAE features. Moreover, especially for cell types, we observe a high *max* selectivity across almost all cell types, while for more complex features we still observe a moderate selectivity score of more than 0.1 across all labels. We present in Table 3 the number of features exhibiting an average selectivity greater than a given threshold for at least one label, across all five classification tasks. This is done using three different thresholds for ICFL with PCA whitening. We observe that dominant concepts, such as cell types, batch effects, and siRNA perturbations that induce strong morphological changes, lead to a substantial portion of features displaying high selectivity. However, also for labels from the functional gene groups (Task 5), we identify more than 100 features with selectivity scores of at least 0.1.

**Separation along feature directions**    The selectivity score analysis showed that activation patterns of the sparse features can be strongly correlated with genetic perturbations. To further strengthen this argument, we illustrate in Figure 3 the cosine similarities between representations from different genetic perturbations and selected feature directions, that is the $i$-th column of $W_{dec}$ for Feature $i$. While we could also directly look at the feature values $z_i$, due to the sparsity, most of the values are 0.

We plot Figure 3 the cosine similarities between selected feature directions and the crop-level aggregated tokens. The histogram in blue represent tokens from specific siRNA perturbations, while the histogram in orange represent all other tokens from Task 3. The plot shows that feature directions effectively separate the two groups, showing that certain features capture important biological information, which shed light on the morphological changes caused by genetic perturbations.

| Threshold | Cell Type | Experiment Batch | siRNA Perturbation | CRISPR Perturbation | Functional Gene Group |
|---|---|---|---|---|---|
| 0.5 | 73 | 11 | 0 | 0 | 0 |
| 0.2 | 455 | 77 | 141 | 2 | 37 |
| 0.1 | 928 | 243 | 681 | 13 | 166 |

Table 3: Feature count (max 8192) with avg selectivity above thresholds for at least one label

## 6.2 COMPARISON WITH FEATURES FROM *CellProfiler*

As a second set of experiments, we compare the average selectivity scores of features from ICFL with those from a set of 964 handcrafted features generated by *CellProfiler* (CP) (Carpenter et al., 2006). These features are designed by domain experts and are widely used for microscopy image analysis. This task compares the monosemanticity of unsupervised features extracted from foundation models to that of human expert-designed features. We obtain sparse features by thresholding the average CP features obtained from all cells from a multi-cell image taken form a subset of the public RxRx1-dataset (Sypetkowski et al., 2023). We threshold at the $\alpha$ and $1 - \alpha$ quantiles with $\alpha$ such that the average number of non-zeros is $\approx 100$. A feature was classified as "activated" when its value, under perturbation conditions, exceeded these quantiles. The selectivities corresponding to both CP and our SAEs, measured using the same datasets.

**Comparison of selectivity scores** In Figure 3e, we plot the highest average selectivity score for each genetic perturbation (a subset of Task 3 in sorted order for both CP features and ICFL features). The results show that the features extracted by ICFL almost match the selectivity scores of the handcrafted, human-designed features. Additionally, in Figure 3f, we show the average score across all labels as a function of various thresholding levels for the CP features. On the $x$-axis, we plot the average number of non-zero elements. We again observe that our features perform comparably to CP features. Interestingly, CP features peak at high levels of non-zeros ($\approx 300$), leaving future work to assess whether this peak selectivity can be matched using deep learning-based approaches while using significantly fewer non-zero elements. We further illustrate the correlation between the best average selectivity scores from the CP and ICFL features for each label (Figure 3g). The plot demonstrates a strong correlation (Pearson coefficient of 0.71), suggesting that ICFL is capable of identifying features that capture patterns similar to those detected by CP.

## 7 QUALITATIVE ANALYSIS OF SELECTED FEATURES

In this section, we illustrate striking, non-trivial patterns captured by selected features and provide an example for how domain experts can interpret, study and validate features found by DL. To study the "semanticity" of features in ViTs, we propose interpreting them at the pixel level by examining which patches exhibit the highest cosine similarities with the feature directions. More precisely, for the multi-cell image crops strongly correlated with selected feature directions, we compute heatmaps of the cosine-similarities of the individual tokens from $8 \times 8$ patches and feature directions (Figure 4 and Figure 5, top rows).

### 7.1 TOKEN-LEVEL FEATURES IN FUNCTIONAL GENE GROUP

We begin by examining a single feature for our interpretabiliy analysis, and its corresponding feature direction, that we chose because it demonstrated a clear biological relationship. The feature is strongly correlated with gene knockouts from the *adherens junctions* pathway, a label from the functional gene perturbation group from Task 5 (§ 5). The adherens junctions connect cell membranes to cytoskeletal elements and form cell-cell adhesions; they can be thought of as "glue proteins" that stick cells together. Our microscopy images visualization in Figure 4 strongly correlated with the feature direction, reflecting this disrupted cellular morphology. The images comprise of small, bright and isolated cells which appear unable to establish proper connections with the neighboring cells (Figure 4, middle row). Note that despite the similar appearance, the images do not originate from the perturbation of a single gene, but rather from a group of genes related in a functional family (Task 5). The regions *most* correlated with the concept direction (Figure 4, top row, most white) belong to areas surrounding perturbed cells. These tokens appear to form a ring-like pattern around the compact cells (top row), which suggests that the concept corresponds to the expected but missing actin (rendered in red) around the cell nucleus (rendered in blue), which is indicative of the perturbation phenotype.

By contrast, the tokens that are *not* aligned with the feature direction associate with cells where the actin meshwork extensively protrudes away from the cell center (yellow bounding boxes). The CRISPR gene editing process (§ 5) is imperfect and as a result in any well, a small proportion of cells remains unperturbed. We found that tokens *least* correlated with the concept direction (Figure 4, top row, most black) belong to what appear to be unperturbed cells (yellow bounding boxes). Examining

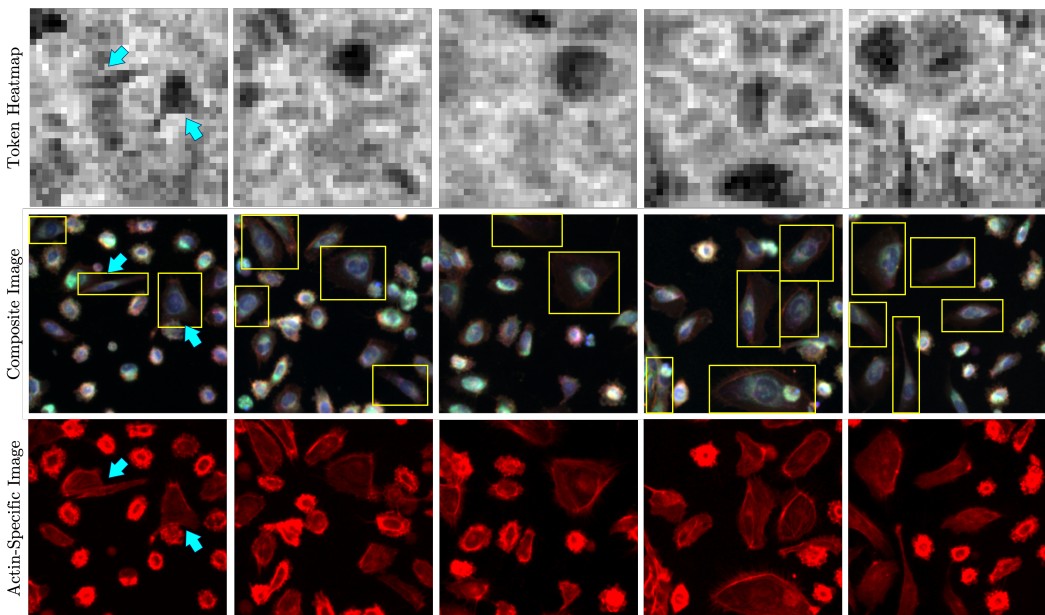

Figure 4: Visualization of composite images (middle row) and their actin-staining channel (bottom row) which strongly correlate with a selected feature from a single functional gene group — *adherens junctions*. Plotted above are the token-level heatmaps of the inner products of the individual tokens with the selected feature direction (top row) for 5 out of 8 strongest correlated images per feature direction. Highlighted are the cells which most likely remain unperturbed (yellow bounding boxes), which are the only instances attempting to establish cell-cell connections (cyan arrows) as they produce the gene to form functional *adherens junctions*.

the corresponding channel-specific image for actin (Figure 4, bottom row) clearly shows that these cells differ from the rest of the well in that they do not contribute to the overall morphology of the image as they manage to form an extensive actin meshwork, and are the only instances which attempt to make connections with neighboring cells (cyan arrows).

## 7.2 CHANNEL-SPECIFIC PROPERTIES OF SINGLE-GENE PERTURBATIONS

Finally, we examine the extent to which we can recover channel-specific signal associated with the gene perturbations. For this exercise, we queried 3 specific gene perturbations: (i) OPA-1, which contributes to the maintenance of correct shape of mitochondria, (ii) ALG-3, which aids in the modification of proteins and lipids in the endoplasmic reticulum (ER) after synthesis, and (iii) TSC-2, which contributes to the control of the cell size (Figure 5).

**OPA-1** The mitochondrial channel shows that most correlated tokens are overlaid with distant regions where enlarged mitochondria are present (pink arrows). Quantitatively, this nuanced relationship does not show a strong correlation in the mitochondrial channel (0.41) due to the aberrant image background, but qualitative examination of the mitochondrial channel highlights this delicate detail which is not obvious from the composite images (Figure 5, $1^{st}$ column, middle row).

**ALG-3** The most aligned tokens appear specific to regions of endoplasmic reticulum (ER) and RNA with which ALG-3 co-localizes, where it aids with attachment of a sugar-like groups to proteins. In this dense image, we report that the correlation of endoplasmic reticulum (0.63) and RNA-specific channels (0.63) are much higher than for channels staining other cellular compartments, *e.g.* plasma membrane (0.24) or actin (0.16). This suggests that our token heatmap is prevalently focused on ER-specific information (Figure 5, $2^{nd}$ column), which is consistent with what we would expect from our understanding of the protein function.

**TSC-2** We examine the plasma membrane- and Golgi apparatus-specific channel to relate perturbed cell size control to the token alignment. We confirm that this channel correlates most strongly with the queried concept direction, but this time in a negative direction. As the plasma membrane — and, hence, cytoplasmic area — are the most extensive from the cell center, the mostly aligned tokens appear to focus specifically on regions which are not covered by the cell membrane, or the membrane

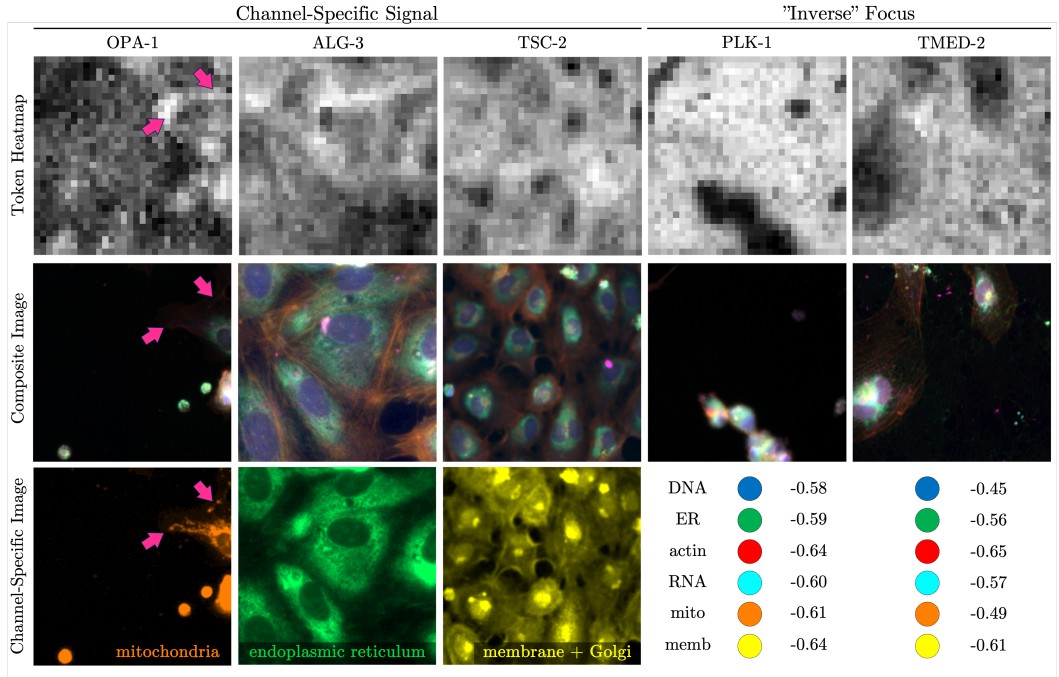

Figure 5: Visualization of representative images from selected single-gene perturbations as in Figure 4 out of 25 strongest correlated images per feature direction. Plotted are the channel-specific staining images of subcellular compartments: mitochondria (orange), endoplasmic reticulum (green) and membrane with Golgi (yellow). Displayed are per-image correlation coefficients between token heatmaps and channel-specific images.

pixel intensity fades away (Figure 5, $3^{rd}$ column, bottom row). This relationship shows highly negative correlation (-0.71), making it a stronger signal than actin (-0.43) or mitochondria (-0.49), and is likely monitoring the lack of channel-specific signal, similarly to the finding from Section 7.1.

**Inverse focus** Finally, we show that the tokens are not always co-localized with regions occupied by cells. Here, we selected two genes which appear to follow an "inverse" trend, namely affecting PLK-1, which enables cell cycle progression through mitosis, and TMED-2, which helps to regulate intracellular protein transport. While both of these gene perturbations render the cells in a characteristic affected state (small, clumped cells struggling to divide *vs.* large, spread out and actively dividing cells), it appears that their most aligned tokens correspond to areas *not* covered by cells, which we confirm with highly negative correlations across all channels (Figure 5, last 2 columns). This suggests that the salient feature for these perturbations is the *lack* of cell density in a well.

## 8 CONCLUSION

In this paper we have explored the extent to which dictionary learning can be used to extract biologically-meaningful concepts from microscopy foundation models. The results are encouraging: with the right approach, we were able to extract sparse features that are associated with distinct and biologically-interpretable morphological traits. That said, these sparse features are clearly incomplete: we see significant drops in their linear-probing performance on tasks that involve more subtle changes in morphology. It is not clear to what extent this is a limitation of our current dictionary learning techniques, the scale of our models, or whether these more subtle changes are simply not represented linearly in embedding space. Nonetheless, it is clear that the choice of dictionary learning algorithm matters to extract meaningful features.

We also proposed a new dictionary learning algorithm, Iterative Codebook Feature Learning (ICFL), and the use of PCA whitening on a control dataset as a form of weak supervision for the feature extraction. In our experiments, we found that both ICFL and PCA significantly improve the selectivity or "monosemanticity" of extracted features, compared to TopK sparse autoencoders. We hope that future work further explores the use of dictionary learning for scientific discovery, as well as the use of ICFL for other modalities like text.

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

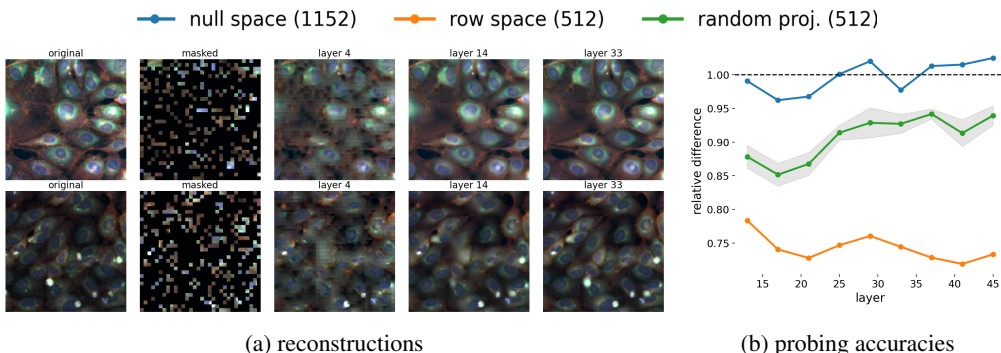

(a) reconstructions           (b) probing accuracies

Figure 6: a) Reconstructions when decoding from intermediate layers. b) The relative linear probing accuracy when using the component from the null space, row space and a random 512-dimensional subspace as component compared to the full component. Both Figures use the MAE-G model.

## A  DISCUSSION: LINEAR CONCEPT DIRECTIONS IN ViT MAEs

We have shown that DL is a powerful approach for finding linear concept directions (features) that are strongly correlated with biological concepts such as cell-types and genetic perturbations. From an interpretability perspective, a question that remains, however, is whether these correlations solely appear due to first order effects of complex non-linear structures used by the model to store abstract information, or whether linear directions are actually inherently meaningful to the model? While linear causal interventions offer strong evidence that the latter may indeed at least be partially true for large language models (see e.g.., (Ferrando et al., 2024) for an overview), there exists relatively little evidence for ViT MAEs besides the high linear probing accuracies on e.g., natural and microscopy image classification tasks Huang et al. (2022); Alkin et al. (2024).

In this section, we provide an argument further supporting the hypothesis that MAEs may rely on linear concept directions when processing data by analyzing at which point in the model are the concepts are the most linearly separable.

**Separation into row- and nullspace.** We note that standard MAE architectures (Huang et al., 2022) use two different embedding dimensions for the encoder block and the decoder block. Both blocks are connected via an *encoder-decoder projection matrix* $W : \mathbb{R}^{d_e \times d_d}$ with, in our case, $d_e = 1664$ (ViT-G model from (Zhai et al., 2022)) and $d_d = 512$. This projection matrix gives raise to a separation of the the tokens into the row-space and null space of $W$, $x = x_{row} + x_{null}$ where only the information stored in $x_{row}$ is passed to the decoder. ViTs and more generally transformer models have shown to align the basis across layers, allowing for decoding of tokens from intermediate layers (Alkin et al., 2024). We visualize this behavior in Figure 6a where we show the reconstructions when using the tokens from intermediate layers. Thus, we observe that the row-space component $x_{row}^l$ of tokens from early and intermediate layers $x^{(l)}$ already store a reconstruction of the masked image that is refined over the layers. Thus the question appears what is the role of the null space component $x_{null}$ which won't be passed to the decoder and thus serves as a "register" (in analogy to Darcet et al. (2023))?

**Component-wise linear probing** We analyze in Figure 6b the different components, showing the relative linear probing accuracy of the probing accuracies using the *null* and *row space* components, compared to the entire token (dashed line at 1) across different layers. As observed, the null space component consistently yields the same probing accuracy as the entire token, while the row space component yields significantly lower accuracy. For comparison, we also show the relative probing accuracy when using a random $d_d$-dimensional subspace (the same dimension as the row space), which consistently yields higher accuracy than that obtained from the row space. These findings suggest that biological concepts (i.e., genetic perturbations) are most linearly separable in the component used only for internal processing during the forward pass and not passed to the decoder, and therefore aligns with the hypothesis that the model represents abstract concepts as linear directions accessed by the layers while processing the data Bricken et al. (2023).

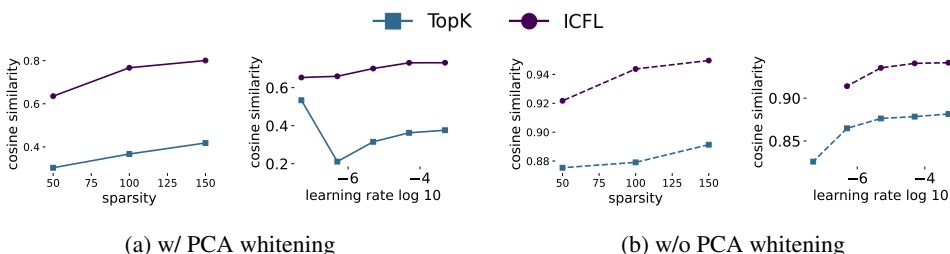

(a) w/ PCA whitening                       (b) w/o PCA whitening

Figure 7: The cosine similarity between the original tokens and the reconstructed tokens for ICFL and TopK-SAE, a) with PCA whitening and b) without, as a function of the sparsity (first and third), i.e. # of non-zeros, and $\log_{10}$ learning rate (second and fourth).

## B INTERPRETABLE FEATURES

In this section, we present additional visualizations of crops strongly correlated with selected feature directions. In the spirit of recent works for LLMs (Bricken et al., 2023), we only present a qualitative analysis that aims to highlight non-trivial, complex, and interpretable patterns captured by these features.

For completeness, Figure 8 shows the same crops as Figure 1 but this time all 6 most correlated and anti-correlated crops. We further present in Figures 9 to 13 additional examples similar to Figure 4 for images strongly correlated with different features. In addition to the heat-map and the entire crop, we also plot the patches that are most strongly correlated with the feature. We make two important observations: a) we can see clear interpretable patterns for which patches are most strongly correlated with the cells, posing a promising area for future research on interpreting and validating concept directions found in large foundation models for microscopy image data; b) we see that the most correlated patches are robust to light artifacts, which can be seen best in the last column in Figure 9.

## C ABLATIONS

In this section we present ablations on type of token, model size, sparsity and learning rate. If not further specified, we always use features extracted from ICFL using PCA whitening.

**Attention block** It is common in the literature to use representations from the MLP output or the attention output (Bricken et al., 2023; Tamkin et al., 2023; Rajamanoharan et al., 2024a). We compare in Table 4 the test balanced accuracy when taking representations from the residual stream and attention output. We observe that both result in similaraccuracies. We make the same observation in Figure 14a and 14b showing an ablation for the linear probes trained on the reconstruction using the same setting as described in Section 6. Moreover, we compare in Figure 15 the selectivity scores as in Figure 2, confirming further that the residual stream and the attention output show a similar behavior. The only exception is TopK for cell types, where the attention outputs result in significantly better selectivity scores, however, still substantially below the ones obtained by ICFL.

| Residual stream | 97.2% | 87.8% | 51.6% | 94.6% | 32.1% |
|---|---|---|---|---|---|
| Attention output | 96.8% | 85.8% | 52.5% | 94.6% | 32.1% |

Table 4: The test bal. acc. for representations taken from the residual stream (Test. Bal. Acc. row from Table 4) and the attention output.

**Model size** We further investigate the model size, as shown in Figures 14a and 14b, where we compare the linear probes for the MAE-G (referred to as Ph2 with 1.9B parameters) with the much smaller model MAE-L (referred to as Ph1 with 330M parameters). We observe that for simple tasks like classifying cell types, both models yield similar performances. However, we observe consistent improvements on complex classification tasks (3,5), both for the probes trained on the original representations, as well as the reconstructions from ICFL and TopK. This demonstrates that dictionary learning benefits from scaling the model size.

We further plot in Figure 16 the selectivity scores. For ICFL, we consistently observe improvements when increasing the model size, while for TopK SAE, we see a significant drop. Interestingly, this drop does not occur for the probing accuracy on the reconstructions in Figures 14a and 14b. This suggests that, although capturing meaningful signals in the reconstructions, TopK SAE faces more difficulties in finding "interpretable" features with high selectivity scores from richer representations post-processed using PCA whitening.

**Sparsity**     As a third ablation, we plot in Figure 7 the cosine similarity of the original tokens and the reconstructed token from the DL for both TopK-SAE and ICFL. We observe that the reconstruction quality of ICFL is much higher than TopK SAE for the same sparsity constraints. This trend persists across all levels of sparsity. The unsupervised reconstruction quality measured by the cosine similarity (or the related $\ell_2$-error) has been often used as a benchmark for SAEs (Rajamanoharan et al., 2024a; Gao et al., 2024).

**Learning rate**     As a last ablation, we also plot in Figure 7 the cosine-similarity for different learning rates. Since PCA whitening leads to more dense tokens, we expect that a decrease in the cosine-similarities, which is also the case when comparing the solid lines (w/o PCA whitening) with the dashed lines (w PCA whitening). Except for TopK-SAE with PCA whitening the reconstruction quality slightly increases with the learning rate (likely due to too few training for small learning rates) and flattens after a learning rate of $5 \times 10^{-5}$, which we choose for all experiments in this paper. Moreover, we observe that TopK-SAE experiences high instabilities when combined with PCA whitening, which is not the case for ICFL.

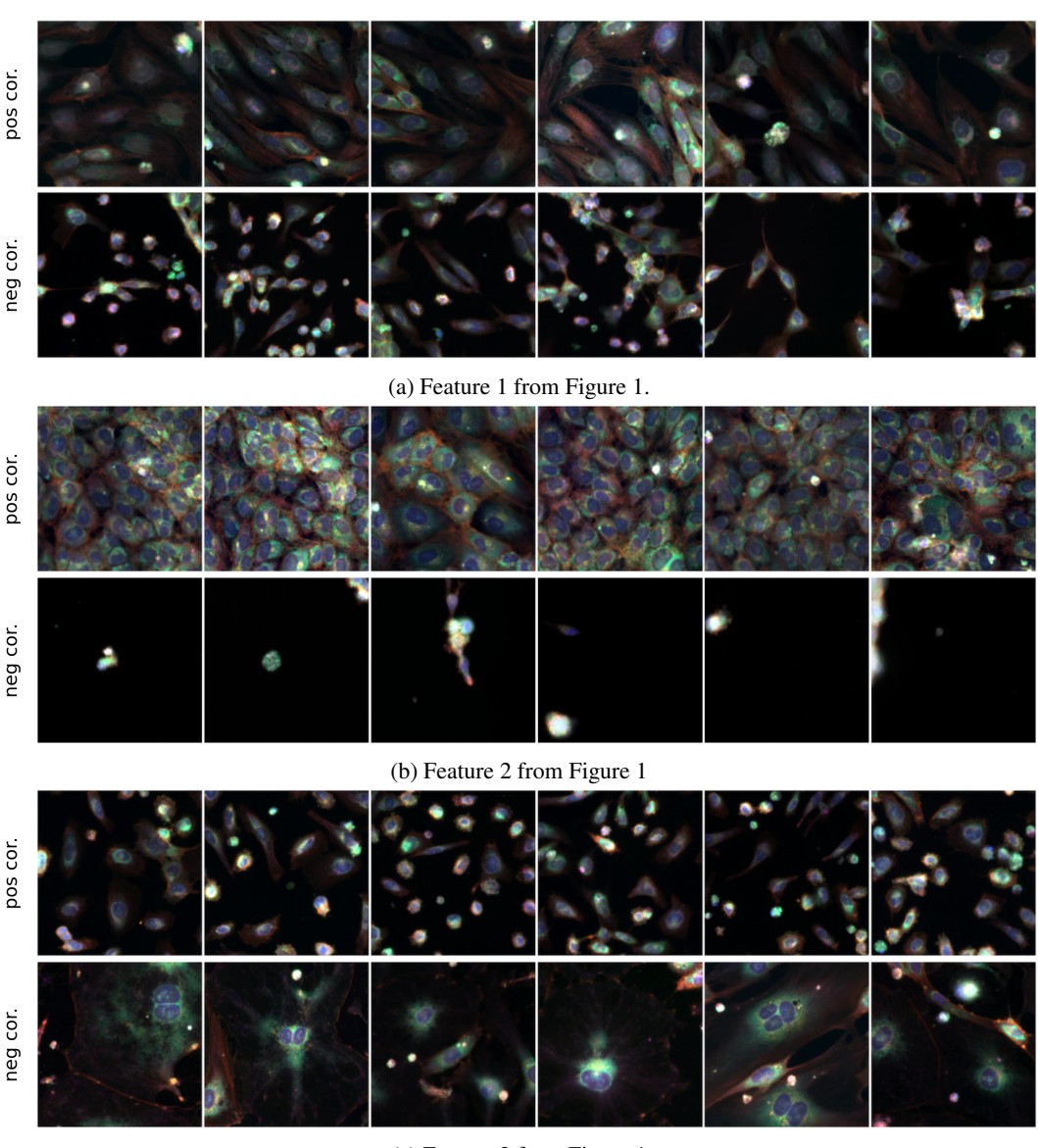

(a) Feature 1 from Figure 1.

(b) Feature 2 from Figure 1

(c) Feature 3 from Figure 1

Figure 8: For each row in Figure 1 we also include the crops that are the most correlated with the feature direction in the opposite direction. More precisely, for each feature we show the 6 most positively (first row) and negatively (second row) correlated crops. For each of the three features we observe a clear concept shift along the feature direction (going from negatively correlated to positively correlated).

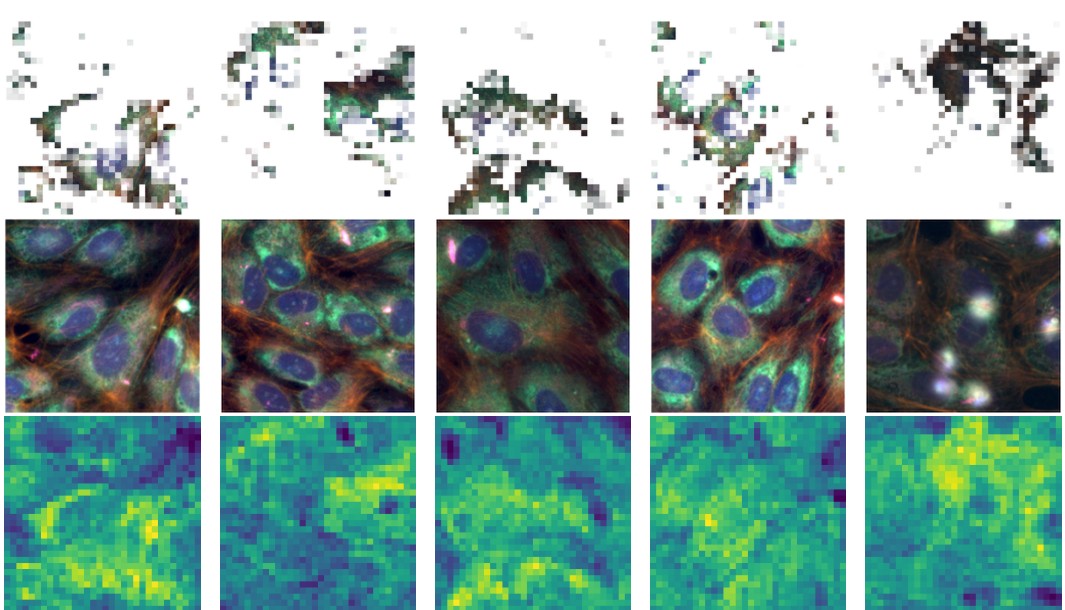

Figure 9: This feature appears to be focusing on the endoplasmic reticuli and nucleoli channel (cyan area) surrounding the nucleus. These are expanded relative to the usual morphology of HUVEC cells.

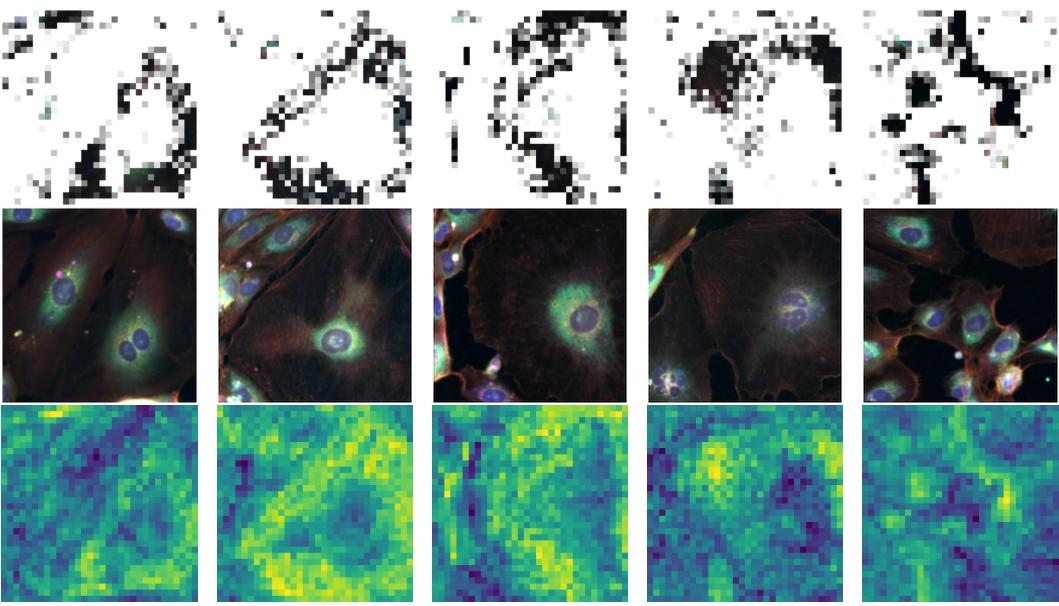

Figure 10: This feature appears to be firing for cells that are unusually large with spread out actin. Note that the feature focuses on the actin channel (red) surrounding the cell.

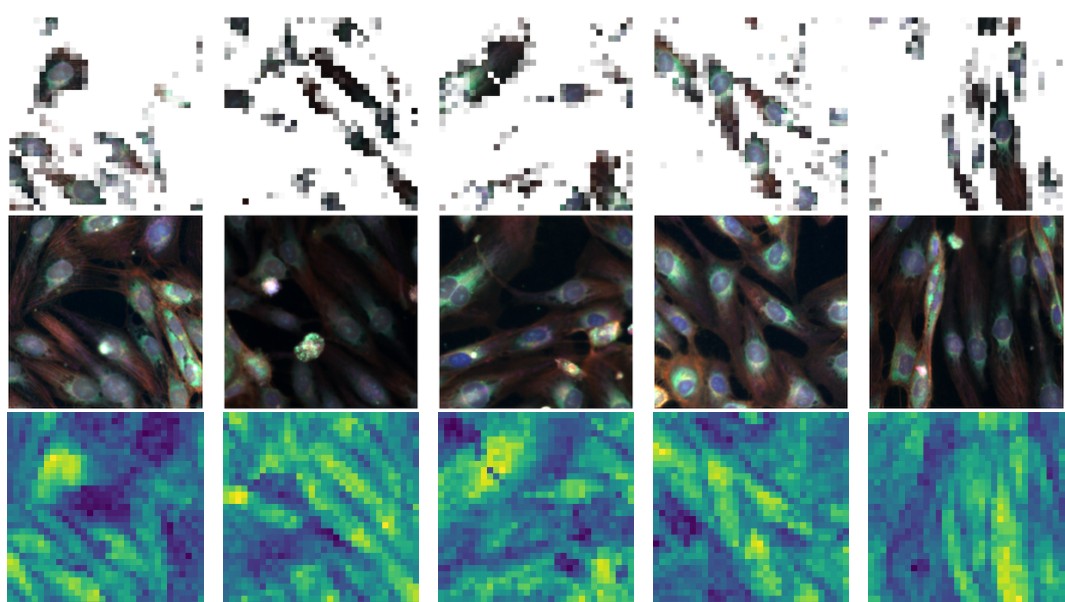

Figure 11: This feature appears to be active for long spindly cells, with the features are most aligned for the long "stretched out" section of the cells.

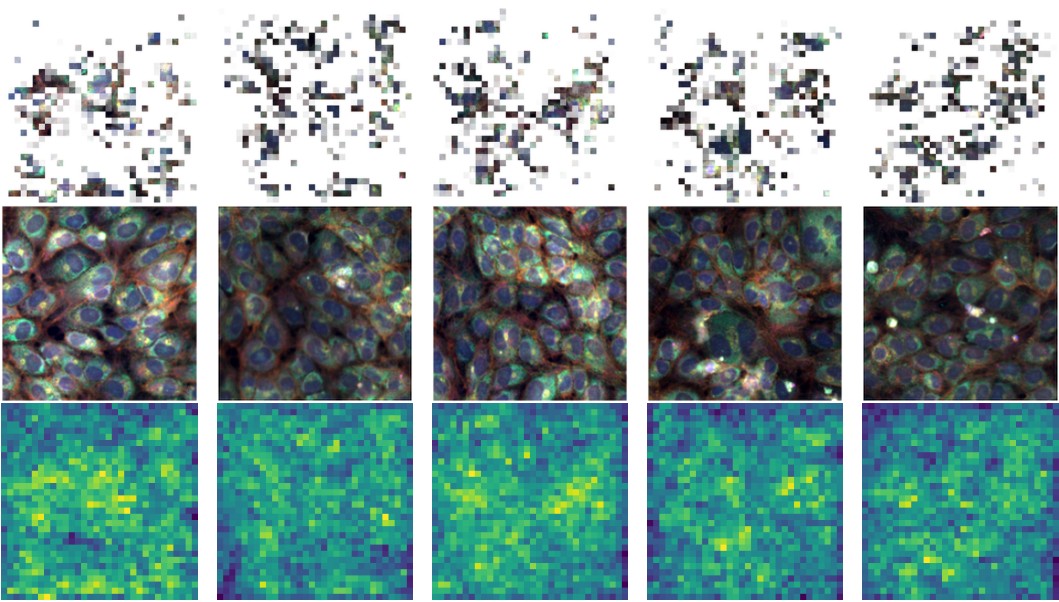

Figure 12: This feature is active for tightly clumped cells. The heatmaps are less clearly interpretable for these images, but appear to be active when neighboring nuclei are touching.

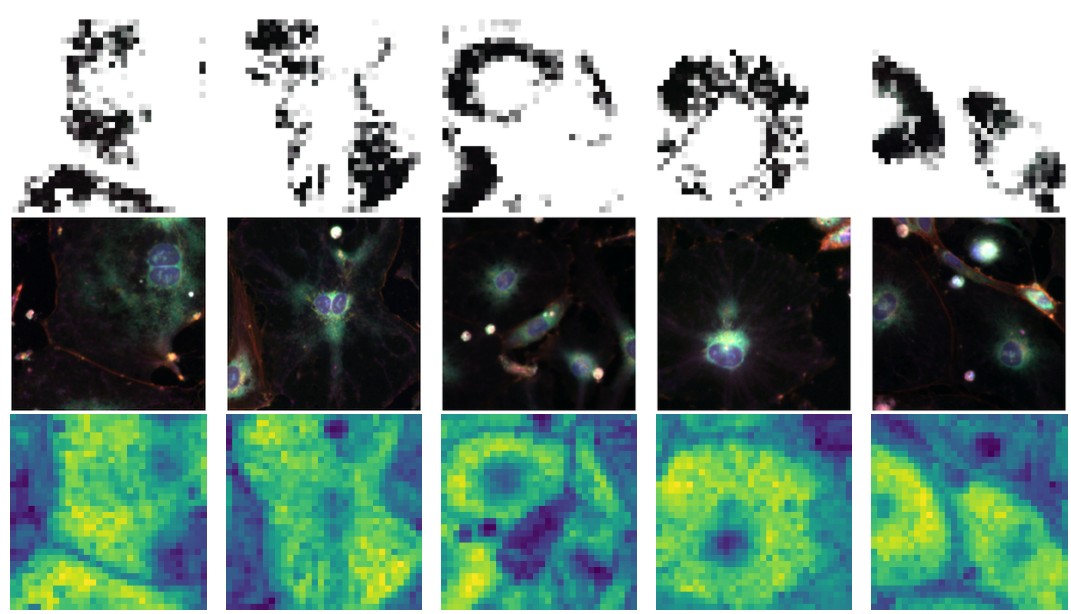

Figure 13: This feature shows a similar behavior to the feature in Figure 10

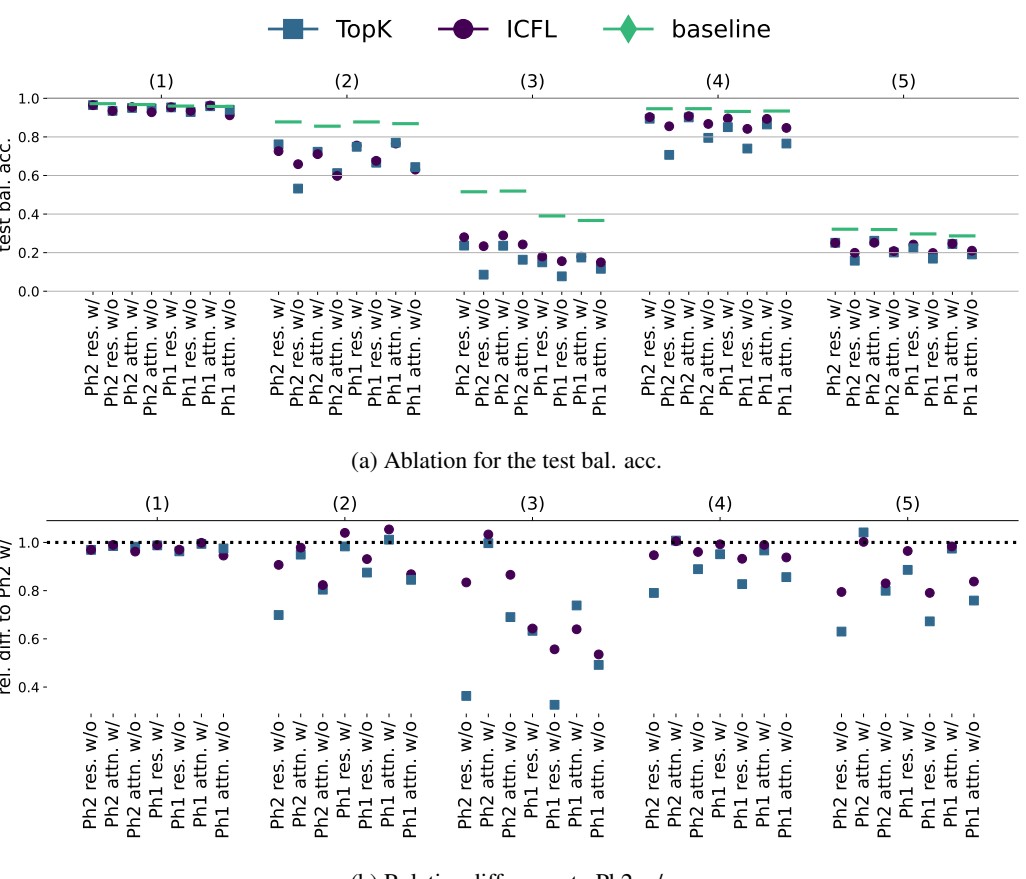

(a) Ablation for the test bal. acc.

(b) Relative difference to Ph2 w/

Figure 14: a) The test bal. acc. of linear probes trained on the original representation (solid lines) and reconstructions from ICFL features and TopK SAEs for representations taken from the residual stream and attention output of Ph2 (larger model) and Ph1 (smaller model), as well as with PCA whitening and without. b) Same as a) but depicting the relative difference in linear probing accuracy compared to Ph2 residual stream using PCA

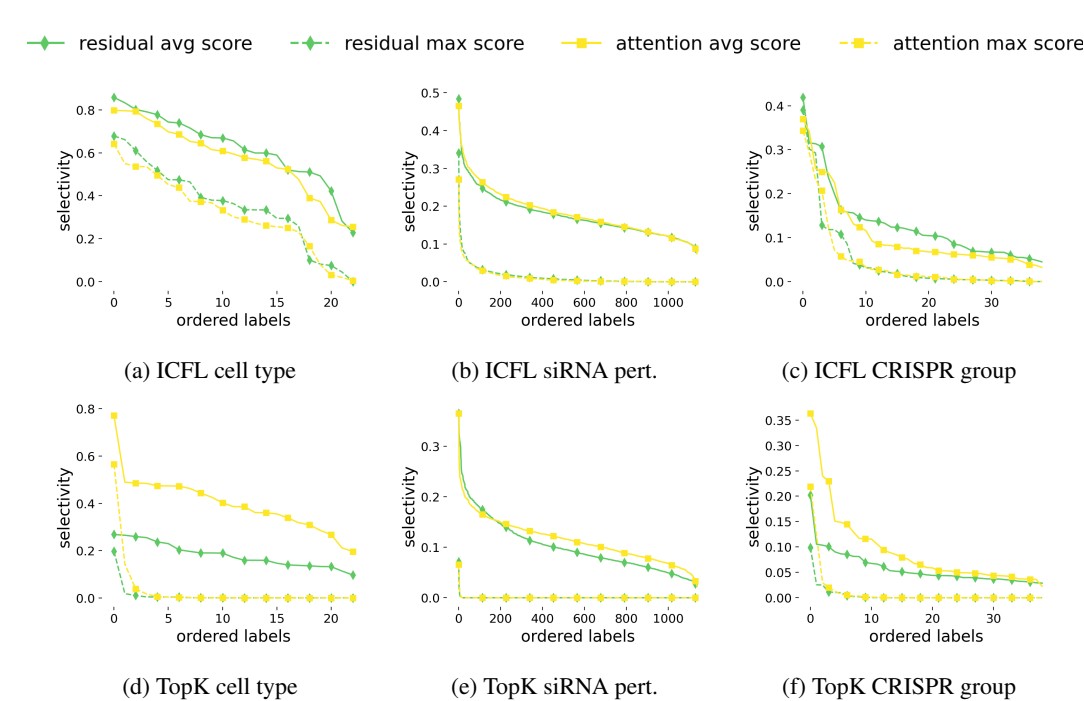

Figure 15: The selectivity scores as in Figure 2 for ICFL (first row) and TopK (second row) when using representations from the residual stream (green) and the attention block (yellow).

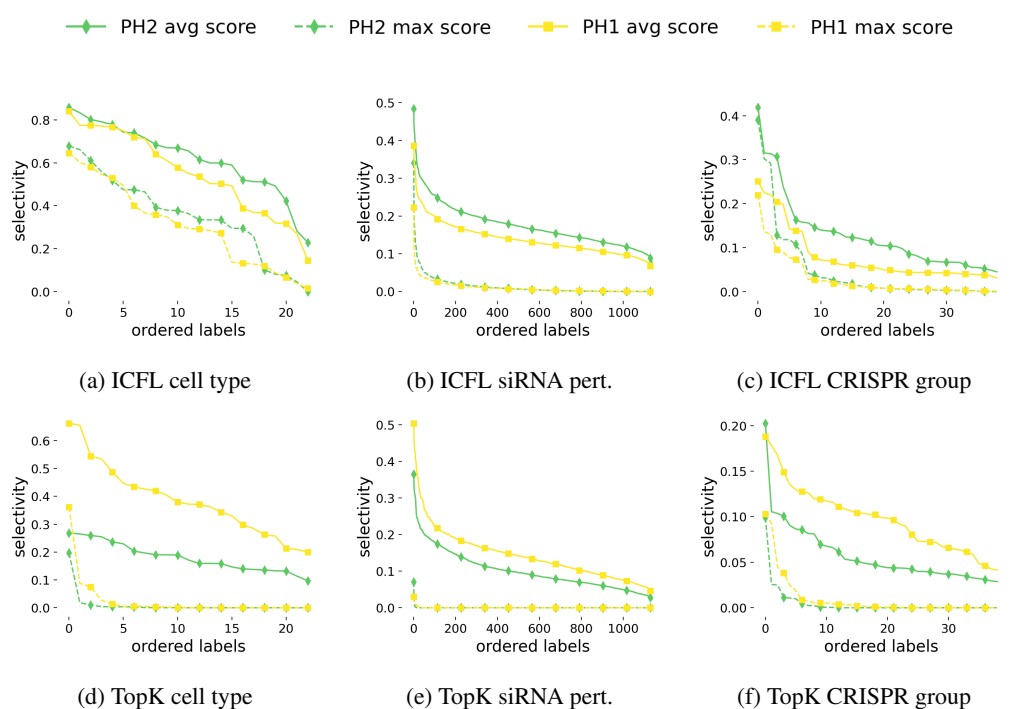

Figure 16: The selectivity scores as in Figure 2 for ICFL (first row) and TopK (second row) when using representations from the residual stream from Ph2 (green) and Ph1 (yellow) using PCA whitening.