# ICLR Rebuttal - Additional Interpretability Results

## 1 Manual single-cell labeling

**Contrasting the tokens to manual single-cell labels:** All 5 images from Figure 5 (main paper), comprising 121 cells in total, were subjected to scoring by a human expert annotator who categorized the cells into 2 subgroups: single cells reflecting the real perturbation or single cells appearing to have escaped the perturbation, which exhibit a control-like phenotype. Out of 31 cells which were scored as control-like by the human annotator, the token level heatmap areas corresponding to these single cells were "darker" in 27 instances. The darker cell areas correspond to tokens with lower alignment to the general direction of the image, and hence are indicative of lower importance of these areas in the image overall. Each cell was only scored as "dark" or "bright" in the token heatmap by an expert annotator after the manual cell labeling was completed. We report that the token heatmaps are capable to "recall" 87% of the expert annotator labels overall (Table 1).

| *Image* | Manual Labeling | | | SAE 'Dark' Cells | Recall |
|---|---|---|---|---|---|
| | **Total Cells** | **Perturbation Cells** | **Control Cells** | | |
| A | 28 | 22 | 6 | 6 | 100.0% |
| B | 34 | 26 | 8 | 6 | 75.0% |
| C | 20 | 17 | 3 | 3 | 100.0% |
| D | 20 | 13 | 7 | 6 | 85.7% |
| E | 19 | 12 | 7 | 6 | 85.7% |
| **Total:** | **121** | **90** | **31** | **27** | **87.1%** |

Table 1: **Comparison of manual single-cell labeling by expert annotator and SAE token heatmap results** for images labeled A–E (as in Figure 5 of the main paper). The "SAE 'Dark' Cells" column represents control-like cells highlighted as 'dark' by the SAE token heatmap. Recall is calculated as the percentage of 'dark' cells over all human-labeled control cells.

**Interpretation of the manual labeling effort:** We report recall and not precision as a metric of choice for this exercise as there exist certain areas which the token heatmaps are explicitly not looking at, such in image C's central left side, as well as image D's bottom left corner. The heatmaps appear to have avoided to focus closely on these regions as they are uninformative—it's hard to distinguish the boundaries of individual cells, and even how many cells are there - there's probably an overlay of cells, with two or more instances growing proximally or even on top of each other.

## 2 Token alignment correspondence to cell types

**Generation of manual segmentation masks:** After visually scoring each single-cell instance by a manual labeling, our expert annotator generated pixel-level segmentation masks of 3 categories to facilitate more robust, quantitative analysis. The image was divided into areas of (i) *black* which corresponds to the image background, (ii) *grey* which reflects cells under the real perturbation, and (iii) *white* which highlights cells

which appear to have escaped the perturbation and are in control-like state. The relative values of the token-level heatmaps were then upscaled to match the shape of the segmentation masks ($256 \times 256$) and (0-1) standardized to allow for comparisons of the heatmaps between images. We computed the distribution of the relative token alignment per each cell class, and compared the difference of their means by one-sided Mann-Whitney U test with significance threshold $\alpha < 0.001$.

**Quantitative analysis of the single-cell areas:** The histograms confirm that the distributions of token alignment for the cell type populations clearly differ between the control-like cells (white) and all other, correctly perturbed single-cell instances (grey), which blend in with the alignment levels similar to the image background (black, Figure 1). In other words, single-cells which appear to have escaped the perturbation phenotype as annotated by human expert, are indicated to be less informative in the overall image than the correctly perturbed cells as well as image background, *i.e.* the presence or absence of the cells, which—at least in the context of *adherens junctions gene knockout*—, is just as informative as the areas occupied by the perturbed cells.

We report that the statistical tests confirm this behaviour and the token alignment of the background and/or perturbation cells are significantly different from the control-like cell populations in all images (Table 2). In images where cell overlay occurs, we report that this is most likely the reason why the mean of the perturbed cells is skewed to the left in the histogram for image "C" and image "D", and why the statistical tests show there's a difference between the tokens occupying background pixels and perturbation-reflecting cells.

| Image | Contrast between categories | *SAE:* $p < 0.001$ | *MAE-G:* $p < 0.001$ |
|---|---|---|---|
| A | Background vs. Perturbed cells | False | True |
| | Background vs. Control-like cells | True | True |
| | Perturbed cells vs. Control-like cells | True | False |
| B | Background vs. Perturbed cells | False | True |
| | Background vs. Control-like cells | True | True |
| | Perturbed cells vs. Control-like cells | True | False |
| C | Background vs. Perturbed cells | True | True |
| | Background vs. Control-like cells | True | True |
| | Perturbed cells vs. Control-like cells | True | False |
| D | Background vs. Perturbed cells | True | True |
| | Background vs. Control-like cells | True | True |
| | Perturbed cells vs. Control-like cells | True | False |
| E | Background vs. Perturbed cells | False | True |
| | Background vs. Control-like cells | True | True |
| | Perturbed cells vs. Control-like cells | True | False |

Table 2: Comparison of SAE and MAE-G results across images and cell categories with p-value threshold.

**Token alignment from naïve MAE-G model:** We compute the same categorical histogram based on the manual segmentation masks, except this time we use the relative alignment of each token to the general direction of the image directly from the *MAE-G* foundation model, without the implementation of our SAE. We express this as the vector dot product between each token representation to the mean across all tokens generated by the *MAE-G* (Figure 2). Here, we can see that the model fails to distinguish between the areas which are important for characterizing this perturbation (i.e. bright perturbation-reflecting cells, low cell density, areas immediately outside the cell boundary, etc.) and pays significantly less importance to the cells which are manually labeled as reflective of the real perturbation.

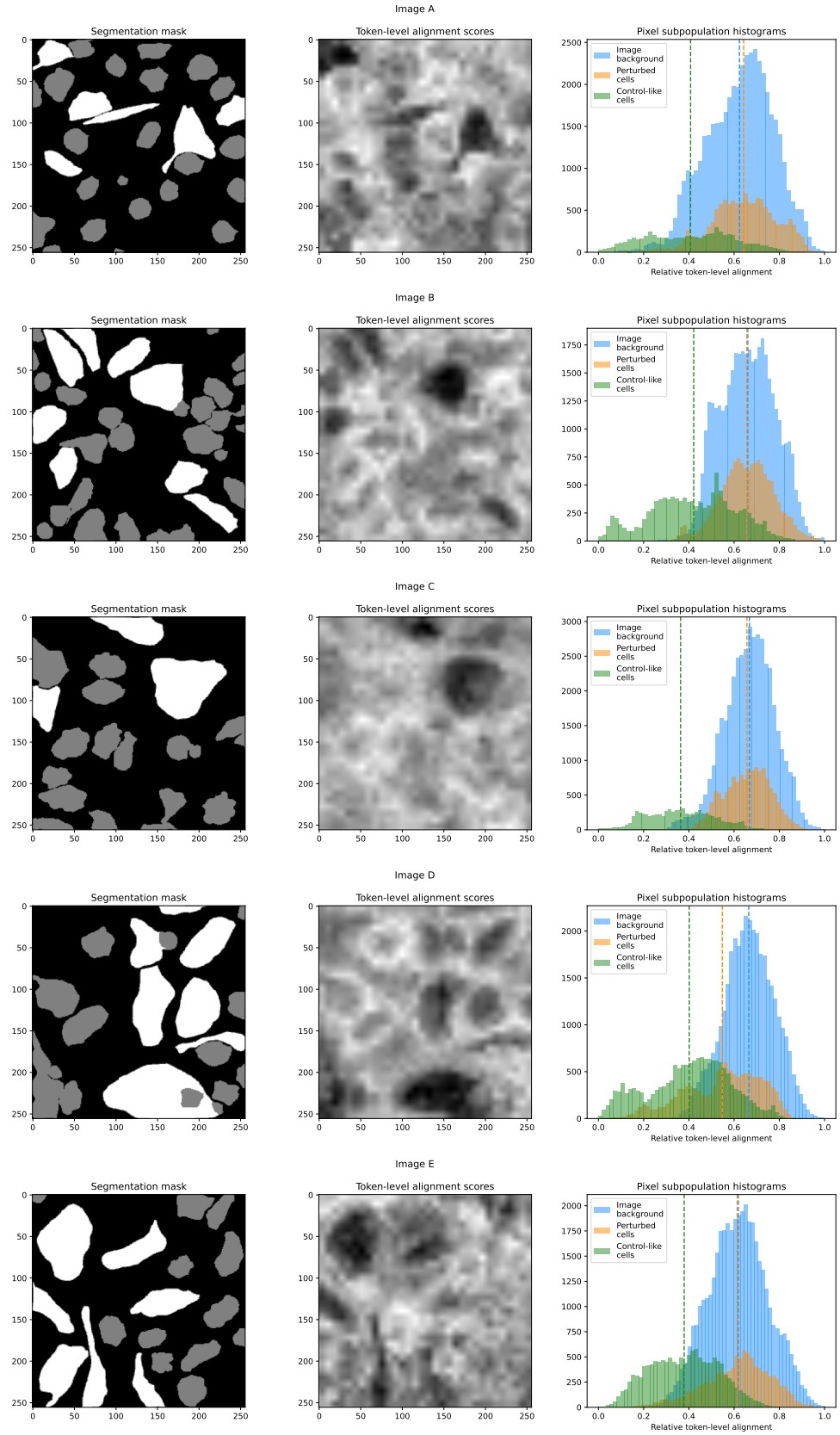

Figure 1: **SAE-generated token-level alignment by cell category** for images *A-E* from Figure 5.

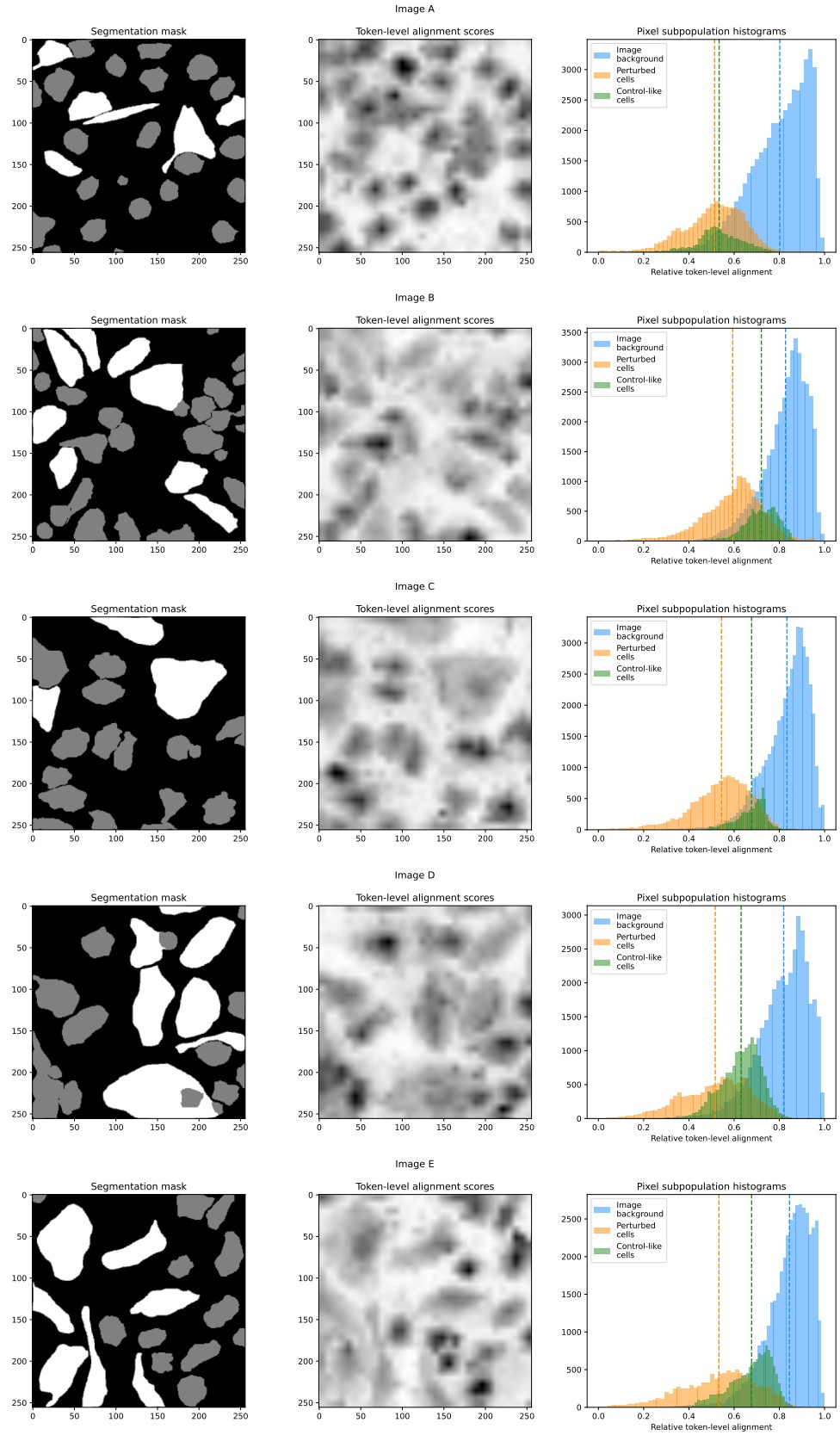

Figure 2: **MAE-G generated token-level alignment by cell category** for images *A-E* from Figure 5.