# OpenReview forum: "Towards scientific discovery with dictionary learning: Extracting biological concepts from microscopy foundation models"
_ICLR.cc/2025/Conference — Submitted to ICLR 2025_

### Official Review · Reviewer_eDB6 · 2024-10-20

**Soundness:** 3
**Presentation:** 4
**Contribution:** 3
**Rating:** 5
**Confidence:** 3

**Summary:**

The paper introduces an algorithm called Iterative Codebook Feature Learning (ICFL). The algorithm consists of a simple linear generator with an l2 reconstruction (as in standard l1 sparse coding) with sparsity being enforced with a topK  (instead of the typical l1) condition on the projections of the data to the decoding matrix columns. The optimal decoding weight matrix is identified with SGD. Once the optimal dictionary is identified the process is iterated with the residual of the image reconstruction for a fixed number of iterations (presumably sufficient to capture the entire structure of the data and set the residual to 0). The ICFL algoirthm is compared against a top-K sparse autoencoder from a recent publication. The comparison looks at the quality of reconstruction and the utility of the full range of dictionary elements suggesting that the ICFL  method yields improved performance. The algorithm is then applied to intermediate level representations from a large masked autoencoder that has been trained on microscopy images, it is described as a foundation model. The authors claim that this enables them to learn sparse features that have physical meaning from foundation models, enabling a new form of scientific discovery.

The sparse coding algorithm seems interesting. However, it seems to me that having a fixed number of iterations would lead to a lot of redundant computation, it would be interesting to see look at the decay of the norm of the residual after ever ICFL iteration. A termination condition based on the norm would seem more sensible to me.
Another interesting direction would be to look at multiresolution features, which are relevant to image structure, usually the iterative algorithms such as those proposed by Mallat & Zhang (1993) lead to features that lie in different parts of the frequency domain and using convolutional operators with dilations to capture structure at different iterations might lead to improved performance of the ICFL algorithm.
While the ICFL demonstrates an interesting approach I am not convinced that studying the internal representation of foundation model can lead to anything particularly meaningful. Why didn’t you work with the original data instead? How can you tell that your result is not a product of some odd averaging (or other form of merging/separation) of features?
I am not a microscopy expert so it is difficult to evaluate that aspect of the paper but I admit that the results seem convincing.
Overall the contribution seems novel and non-trivial, however, its future scientific significance appears to be vague to me at this point.

**Strengths:**

It is a non-trivial technical contribution. The paper appears to combine expertise from multiple areas including at least foundation models in microscopy, cellular microscopy, and sparse coding. There is a novel method of analysis of foundation models that might lead to a significant contribution to cellular microscopy.

**Weaknesses:**

I don't see the point of analysing the intermediate layers of a foundation model for domain specific scientific reasoning. It makes sense in trying to undestand what foundation models do exactly but that does not seem to be the goal here. I am not sure I fully understand the main ambition of the paper.

**Questions:**

What features do you get when you apply ICFL directly on cellular microscopy images?
Can you provide a plot that demonstrates the decay of the norm of the image residual after successive ICFL iteration?
Can you elaborate on the significance of the obtained features?
Why did the token heatmap specialise on the missing actin (line 426)?
Can you give more details on the foundation model (e.g. input, output)?
What do you think your model can learn from a foundation model, something related to its input or its output, or both?

---

> ### Author Response · Authors · 2024-11-21
> **Response to Reviewer eDB6**
>
> **Motivation for studying intermediate layers:**
>
> >I don't see the point of analysing the intermediate layers of a foundation model for domain specific scientific reasoning. It makes sense in trying to undestand what foundation models do exactly but that does not seem to be the goal here. I am not sure I fully understand the main ambition of the paper.
> > Can you elaborate on the significance of the obtained features?
>
> We would like to refer the reviewer to the section **Positioning of the paper** in the response addressing all reviewers and may ask them whether this clarified their concern. We specifically chose intermediate layers over the final layer because we aim to extract features from the point in the residual stream where the model retains the most biological information. For MAEs, we observe that the peak information is located in an intermediate layer. Additionally, using intermediate layers is a common practice in mechanistic interpretability research on large language models (LLMs) as noted in previous studies ([1,2]).
>
>
> **Scientific significance:**
>
> > Overall the contribution seems novel and non-trivial, however, its future scientific significance appears to be vague to me at this point.
>
> We would like to refer the reviewer to the section **Scientific relevance** in the common response to all reviewers.
>
> ### Questions:
>
> **More computationally efficient variants of ICFL:**
> > A termination condition based on the norm would seem more sensible to me.
>
> We agree that future research in this direction would be interesting! Since compute is not a limiting factor in comparison with the required compute of training the foundation models, we did not explore more computationally efficient methods yet. However, when scaling DL compute certainly becomes a valid concern and  techniques that adaptively choose the number of steps could be a promising solution.
>
>
> **More computationally efficient variants of ICFL:**
> >  Can you elaborate on the significance of the obtained features?
>
> We thank the reviewer for raising this important question. We would like to refer the reviewer to Section **Scientific relevance** in the common response addressing all reviewers.
>
>
> **Features extracted directly from the microscopy image data:**
> > What features do you get when you apply ICFL directly on cellular microscopy images?
>
> May we ask the reviewer to specify their question? Is the reviewer suggesting applying ICLF to the flattened image containing RGB values? If this is the case, we do not expect any meaningful features as ICFL only extracts linear directions. Thus, ICFL can only be used in combination with a large scale (transformer) model, in strong contrast to disentangled representation learning methods mentioned in the related works.
>
>
> **Features extracted from the foundation model:**
> >  Can you give more details on the foundation model (e.g. input, output)?
>
> The foundation model is a gigantic masked auto-encoder (MAE-G) based on a vision transformer (ViT) architecture that takes multi-channel images as input into the encoder and returns the reconstructed version of the image of an identical shape as an output of the decoder. In the process of the image encoding, token-wise embeddings are produced at masking ratio of 0 at inference time.
>
> > What do you think your model can learn from a foundation model, something related to its input or its output, or both?
>
> By extracting features from the intermediate representations of the MAE, we hope to learn abstract features capturing biologically relevant information about the underlying input data. We would like to refer the reviewer also to the section **Positioning of the paper** in the common response to all reviewers.
>
>
>
> [1] Gao, Leo, et al. "Scaling and evaluating sparse autoencoders." arXiv preprint arXiv:2406.04093 (2024).
>
> [2] Bricken, Trenton, et al. "Towards monosemanticity: Decomposing language models with dictionary learning." Transformer Circuits Thread 2 (2023).

---

> > ### Author Response · Authors · 2024-11-24
> > **Follow-up**
> >
> > Dear Reviewer eDB6,
> >
> > We hope our rebuttal has addressed your concerns and answered your questions. As the end of the discussion period approaches, we would like to ask if you have any additional questions or concerns, particularly regarding our rebuttal.
> >
> > Thank you once again for your time and thoughtful feedback!

---

> > > ### Comment · Reviewer_eDB6 · 2024-11-25
> > >
> > > Thank you for the time and effort you dedicated towards answering my questions.
> > > My greatest hesitation regarding your work has to do with the scientific relevance.
> > > I understand your motivation as it relates to the discriminative nature of ML features; I do not understand the motivation from a biology/microscopy perspective.
> > > What do biologists learn if they know that one of your features emerges in the hidden layers of a foundation model?
> > > It might be something significant or something insignificant. It gains some significance in that it is a more discriminative features than most others, but it is discriminative through an obscure (black-box) decision process, so one could argue "who cares? that's not how I make decisions".
> > > As an ML expert, and not a biology/microscopy expert,  I would like to be optimistic and say that this might lead to something useful and accept the publication of this method at ICLR. However, I think it would be more fair to recommend that you elucidate the scientific value of your new methodology and try again, perhaps in a venue more relevant to microscopy?

---

> > > > ### Author Response · Authors · 2024-11-26
> > > > **Response to Reviewer eDB6**
> > > >
> > > > We thank the reviewer for their feedback and for raising an important point about the scientific relevance of our work from a biology/microscopy perspective. Below, we address this concern by clarifying the biological motivation and impact of our methodology.
> > > >
> > > > Large-scale drug discovery screening campaigns (e.g., the Drug Discovery Unit at Dundee [1], JUMP-CP [2], RxRx1/RxRx3 datasets [3, 4]) are increasingly reliant on machine learning for quantifying similarities between perturbations. However, these approaches often lack interpretability, leaving open questions about why certain perturbations are similar and whether the underlying signals are biologically meaningful or driven by confounding factors, such as batch effects.
> > > >
> > > > Our methodology addresses this critical gap by elucidating the biological basis to the image representation relationships. By identifying emergent features that align with single-cell perturbation patterns, we provide biologists with actionable insights into the mechanisms of action of compounds or gene pathways. This enables a deeper understanding of cell morphological complexity, going beyond black-box predictions to facilitate informed decision-making.
> > > >
> > > > A unique strength of our approach lies in its ability to identify specific single-cell instances that align most strongly with emergent features. This allows biologists to assess shared patterns across perturbations and distinguish meaningful signals from noise. For example, in our experiments (Figures 4 and 5), we demonstrate how the method highlights perturbed cells without prior annotations, showcasing its utility in real-world settings where manual annotation is infeasible.
> > > >
> > > > Manual annotation of datasets, such as those in our study (e.g., up to 30 cells per image, Figure 4), takes ~30 minutes per image set for an expert. Scaling this process to millions of images is impractical. Our method automates this workflow, significantly reducing time and labor while maintaining biological interpretability. This positions our work as a key enabler for advancing single-cell method development and scaling drug discovery pipelines.
> > > >
> > > > We appreciate the reviewer’s optimism and agree that elucidating the scientific value of our method is essential. We believe our work addresses a critical need in the field by bridging the gap between machine learning predictions and biological interpretability. While further work to scale our approach is ongoing, we are confident that this contribution represents a meaningful advance with broad potential impact in microscopy and beyond.
> > > >
> > > > [1] Drug Discovery Unit, University of Dundee. (2024) Retrieved November 26, 2024, from https://drugdiscovery.dundee.ac.uk/
> > > >
> > > > [2] Chandrasekaran, S. N., et al. JUMP Cell Painting dataset: morphological impact of 136,000 chemical and genetic perturbations. (2023) bioRxiv 2023.03.23.534023; doi: https://doi.org/10.1101/2023.03.23.534023
> > > >
> > > > [3] Sypetkowski, M., et al. RxRx1: A Dataset for Evaluating Experimental Batch Correction Methods. (2023) arXiv. https://doi.org/10.48550/arXiv.2301.05768
> > > >
> > > > [4] Fay, M. M., et al. RxRx3: Phenomics Map of Biology. (2023) bioRxiv. https://doi.org/10.1101/2023.02.07.527350

---

### Official Review · Reviewer_2RjZ · 2024-10-29

**Soundness:** 3
**Presentation:** 3
**Contribution:** 2
**Rating:** 5
**Confidence:** 4

**Summary:**

This paper proposes a method for dictionary learning on microscopy images to extract concepts from the image data in an unsupervised fashion.  The proposed method, Iterative Codebook Feature Learning, is an alternative to TopK sparse auto-encoder, that uses an orthogonal matching pursuit approach, resulting in fewer unused/"dead" features.  Experimental results confirm that the learned representations with ICFL have fewer dead features relative to TopK, and improved selectivity with respect to labels from several classification tasks.  Some qualitative analysis is given of the learned features.

**Strengths:**

+ The paper studies an interesting and important domain, where the question of whether meaningful concepts can be extracted automatically from data has significance to biological science.

+ Empirical results are generally well-organized and support the claimed benefits of using ICFL over TopK, specifically with respect to improved selectivity scores and reconstruction quality.  Results also highlight the importance of PCA whitening as a pre-processing step.  Comparison is also given with handcrafted features, and the paper is clear on limitations/areas where current methods fall short of desired performance.

**Weaknesses:**

- With respect to machine learning methodology, the technical novelty of the paper is somewhat limited.  The overall problem and approach follows TopK, except rather than optimizing the specific TopK formulation, an orthogonal matching pursuit approach is used instead.

- See also questions below.

**Questions:**

- ICFL was shown to be superior to TopK on some quantitative measures, but these measures are proxies and not an end goal in themselves.  Qualitative results were only given for ICFL.  Was any qualitative analysis done on the TopK learned features?  Was it the case that those features were clearly sub-optimal to ICFL from a qualitative perspective?  Or were they also able to capture any concepts like the morphologies shown in Figure 1?

- The qualitative analysis gives a discussion and intuition for a small number of the learned features.  Was it the case that many of the learned features had this kind of interpretability as human-understandable concepts?  Or was this only true for the small subset shown?

- More generally, how do you see the results of this method being used in biological research?  Is the desired goal that by looking through the learned features, a researcher might find a single concept that captures something complex and unexpected, warranting further study?  I ask this in regard to the overall positioning and motivation of the paper as discussed in the related work.  For instance, from the perspective of biological research and understanding, why not instead take a CRL approach that directly tries to learn interpretable features/representation?  Or if certain tasks are of key importance, why not focus on learning concept vectors with respect to that task/class-labeled data.  It would be interesting to compare the proposed method with class-based concepts using something like Ghorbani et al Automatic Concept-based Explanations, for instance using the specific tasks already being considered in the paper.

---

> ### Author Response · Authors · 2024-11-20
> **Response to Reviewer 2RjZ**
>
> **Technical novelty:**
>
> > With respect to machine learning methodology, the technical novelty of the paper is somewhat limited. The overall problem and approach follows TopK, except rather than optimizing the specific TopK formulation, an orthogonal matching pursuit approach is used instead.
> >ICFL was shown to be superior to TopK on some quantitative measures, but these measures are proxies and not an end goal in themselves. Qualitative results were only given for ICFL. Was any qualitative analysis done on the TopK learned features? Was it the case that those features were clearly sub-optimal to ICFL from a qualitative perspective? Or were they also able to capture any concepts like the morphologies shown in Figure 1?
>
> We would like to refer the reviewer to the section **Novelty of the ICFL algorithm** in the common response to all reviewers.
>
> ### Questions:
>
> **Number of interpretable features:**
>
> >The qualitative analysis gives a discussion and intuition for a small number of the learned features. Was it the case that many of the learned features had this kind of interpretability as human-understandable concepts? Or was this only true for the small subset shown?
>
> The reviewer raises an important question. While our qualitative analysis suggests that many of the features capture at least some biologically meaningful information, expanding this analysis poses challenges. We extend our qualitative analysis in the section **Interpretability Analysis** in the common response to all reviewers. However, we would like to stress that, in its current form, we cannot provide a definitive answer to the reviewer's question, as addressing it would require significant effort. In particular, it is non-trivial for a scientist to determine, by examining examples, whether a randomly chosen feature captures potentially complex biological patterns or simply overfits to batch effects. Developing protocols that involve domain experts to assess the quality of SAEs and to facilitate the discovery of features capturing novel, biologically relevant patterns represents a very important and promising interdisciplinary direction for future research.
>
>
> **Application in biological research:**
>
> >More generally, how do you see the results of this method being used in biological research? Is the desired goal that by looking through the learned features, a researcher might find a single concept that captures something complex and unexpected, warranting further study?
>
> We thank the reviewer for raising this important question. Yes, this is one of the potential applications. We would like to refer the reviewer to section **Scientific relevance** in the common response to all reviewers.
>
> **Alternative approaches:**
> >For instance, from the perspective of biological research and understanding, why not instead take a CRL approach that directly tries to learn interpretable features/representation? Or if certain tasks are of key importance, why not focus on learning concept vectors with respect to that task/class-labeled data. It would be interesting to compare the proposed method with class-based concepts using something like Ghorbani et al Automatic Concept-based Explanations, for instance using the specific tasks already being considered in the paper.
>
> We thank the reviewer for raising this important question. We mention alternative approaches in the related work and will extend this section in a revised version of the paper. To answer the reviewers question, we would like to mention the following two aspects:
>
> *Supervised vs unsupervised features:* Supervised features are expected to be more predictive of the labels and help researchers better understand which patterns separate classes. However, multiple factors limit supervised feature learning. First, off-target effects of the CRISPR guides are significant  [Lazar et al 2024, “Proximity bias”], and so there is a very real risk that the features that are most predictive of a given target, are those that depend on cells experiencing off target effects rather than the phenotype of interest.  In this context, we point the reviewer to recent studies that demonstrate the superiority of biological representations learned through self-supervised methods over supervised methods, particularly when evaluating the recall of known biological genetic interactions.
>
>
> *Large scale foundation models:* In this paper we extract features from self-supervised vision models that can be efficiently scaled and trained on large scale datasets. In contrast, many works on disentangled representation learning usually learn an auto-encoder from scratch, drastically limiting their applicability on large scale datasets.

---

> > ### Author Response · Authors · 2024-11-24
> > **Follow-up**
> >
> > Dear Reviewer 2RjZ,
> >
> > We hope our rebuttal has addressed your concerns and answered your questions. As the end of the discussion period approaches, we would like to ask if you have any additional questions or concerns, particularly regarding our rebuttal.
> >
> > Thank you once again for your time and thoughtful feedback!

---

> > > ### Comment · Reviewer_2RjZ · 2024-11-25
> > >
> > > Thank you for your detailed response.  Reading over all reviews and feedback, from a high level perspective, it seems two primary concerns are with the technical novelty of the paper, and the overall positioning/scientific significance.
> > >
> > > My assessment, taking the reviews and feedback into account, is that the paper shows potential in the particular proposed direction.  However, the results seem more suggestive of or proxies for the ultimate desired goals.  In its current state, then, I think it may be below the threshold for a general conference like ICLR, and perhaps better suited for a more domain-specific conference.  Pushing past this threshold I think could potentially be done in one of several ways - improving/expanding the core method, showing broader applicability to other related domains, or strengthening the results to make a stronger case for the significance.

---

> > > > ### Author Response · Authors · 2024-11-26
> > > > **Response to Reviewer 2RjZ**
> > > >
> > > > We thank the reviewer for their feedback. While we understand the concerns raised, we present a different perspective on the paper’s overall positioning and scientific relevance.
> > > >
> > > > Our work focuses on learning single-cell heterogeneity in multi-cell imaging data, with demonstrated discriminative power to identify perturbed and non-perturbed cells. In the six gene perturbations we chose for demonstration, our method consistently uncovers biologically meaningful patterns. Specifically, it interprets the source of cell heterogeneity and perturbation phenotypes on both the single-cell (Figure 4) and channel-specific (Figure 5) levels. These findings go beyond being proxies, instead offering direct, actionable insights into biological processes.
> > > >
> > > > Furthermore, our approach expands beyond traditional image classification tasks. It enables interpretable analysis in single-cell imaging by aligning correctly perturbed cell instances with key feature directions, without relying on prior knowledge of channel composition or cell annotation. This interpretability underscores the scientific significance and novelty of our method, distinguishing it from existing approaches.
> > > >
> > > > While scaling to a broader range of gene perturbations remains future work, the current method already provides substantial value. For instance, manual annotation of a dataset like ours (e.g. 130 cells in Figure 4) takes approximately 30 minutes per 5 images by an expert human annotator. This process is infeasible for datasets comprising millions of cells. By automating this step, even partially, we significantly advance the field by enabling single-cell method development at scale.
> > > >
> > > > We believe our work lays a strong foundation for further advances in single-cell image analysis. Its potential to bridge gaps in automated annotation and representation learning aligns well with the interests of the broader computational biology and machine learning communities. We are confident that this work represents a meaningful and impactful contribution to the field.

---

### Official Review · Reviewer_o17p · 2024-11-02

**Soundness:** 3
**Presentation:** 2
**Contribution:** 2
**Rating:** 5
**Confidence:** 3

**Summary:**

The paper focuses on sparse dictionary learning for unsupervised learning of concepts from microscopy images. The target vectors are embeddings from microscopy foundation models and the dictionary is overcomplete, intended for sparse combination of latent concepts to reconstruct the target embeddings. The proposed method ICFL closely resembles OMP for L0 sparse recovery. The authors perform in-depth analyses to demonstrate the strengths of their approach for microscopic images.

**Strengths:**

This is an interesting study where the authors combine classical signal processing idea of dictionary learning (from 2006 when K-SVD was first proposed) and recent foundation models for microscopy images. This shows that classical ideas can feed off from the latest advancements in deep learning to promote the interpretability of the results, which deep learning (despite all the recent saliency methods) clearly lacks. The authors perform several in-depth ablations, both quantitatively and qualitatively, to demonstrate that the learned latent concepts 1) do not lose the "downstream" performance, such as cell type classification and 2) also enable interpretability analysis of each learned concept.

**Weaknesses:**

While I believe the study is thorough as explained above, there are few concerns that I think are holding back the paper.

1. **Presentation** I think the authors are presenting the paper in very verbose manner, which made it confusing to get main messages from the paper. For instance, I am not sure if the "Superposition hypothesis" and "disentanglement and causal representation literature" were necessary to lay out the logic for the paper - The authors could have simply dived in to why dictionary learning is desirable.

Also for explaining results, I sometimes found myself lost in why each of the experiment is important and how it ties to the central message of ICFL (e.g., Fig.2b, Fig.2C, Fig.3f, Fig.3g). Also the authors in the results section switch between different tasks (among the five tasks and experiments) without sufficiently explaining, and this made it hard to follow the paper. I would like the authors to really work on the clarify of the paper.

2. **Technical novelty** At the end of the day, ICFL is a variation of Orthogonal Matching Pursuit and also restricted to L0 recovery methods, which has been around since early 1990s. The added novelty could be that the target is an embedding vector, but this wouldn't really considered a technical novelty typically required for ICLR submissions. For ICFL to really be a great contribution, I think the algorithm should also be able to handle more-widely used L1 recovery methods, given plethora of works in this space (both in classical signal processing sense and more recent neural network sense, through unrolled networks for sparse recovery)

I also think for this to be considered as a robust concept recovery method, additional experiments are required for other variations of microscopy FMs as well. The authors restrict the analyses on MAE-pretrained ViTs, whereas if the authors also perform similar analyses on other types of SSL-pretrained ViTs for this domain, I think that would strengthen the argument that ICFL is robust.

3. **Lack of takeaway messages \& Baselines** The authors largely compare ICFL against two major baselines. Top-K SAE and CellProfiler. It is great that it is performing better than Top-K SAE, which I consider a lower bound, but this doesn't seem to tell you much about the performance one would hope to reach for actual downstream tasks \& Biomarker discovery research. The reason is that ICFL lags behind on almost all metrics behind the original embedding baseline and CellProfiler-derived baselines - As a reader/possible user of the algorithm, I find it less convincing to use ICFL, if it is always performing worse than simpler and older approaches.

I think the authors try to counterbalance by devoting lots of effort in to qualitative analysis part, but due to the presentation issues outline above, I found it less convincing. As an example of lack of takeaway messages, in Table 3, it's good that for lower threshold you get some degree of features with good selectivity, but what does that really mean in terms of using it for actual research? What does Cell Type 928 feature count with Threshold 0.1 mean in this context? How should a scientist interpret this?

4. **Minor points**
- Figure 3 caption. No description of (e), (f), (g) - I think b) c) were intended for that.
- I am not sure what Pearson correlation of 0.71 is supposed to mean for CP and ICFL features. Doesn't this mean certain labels are just more likely to be activated than other labels? Not necessarily that ICFL is capable of identifying the features similar to CP.
- I think the authors need to experiment with different size of dictionary. Why 8,192 only?

**Questions:**

- In line 319, why use center crop, instead of the whole image?

---

> ### Author Response · Authors · 2024-11-20
> **Response to Reviewer o17p**
>
> **Writing style:**
> > The authors could have simply dived in to why dictionary learning is desirable.
>
> While we appreciate the reviewer's constructive feedback regarding the paper's writing, we would like to emphasize that our motivation for using deep learning stems directly from the superposition hypothesis. Furthermore, the field of causal representation learning addresses a similar problem to this paper but adopts a very different approach, and indeed some papers have already begun to make formal connections between the two areas [ Rajendran et al 2024, “Learning interpretable concepts: Unifying causal representation learning and foundation models”].
>
> **Contribution of individual experiments to the story line:**
> > Also for explaining results, I sometimes found myself lost in why each of the experiment is important and how it ties to the central message of ICFL (e.g., Fig.2b, Fig.2C, Fig.3f, Fig.3g).
>
> We would like to refer the reviewer to the paragraphs **Positioning of the paper**  and Novelty of the **Novelty of the ICFL algorithm**
>  in the comment addressing all reviewers.
>
> **Switching between different classification tasks in experiments:**
> >Also the authors in the results section switch between different tasks (among the five tasks and experiments) without sufficiently explaining, and this made it hard to follow the paper.
>
> We agree with the reviewer that switching between different tasks can interrupt the flow while reading the paper. We tried to always select the tasks that are most relevant for the specific experiment and we will add additional explanations to better guide the reader through the experimental section.
>
>
> **Technical novelty of the ICFL algorithm:**
> > Technical novelty At the end of the day, ICFL is a variation of Orthogonal Matching Pursuit and also restricted to L0 recovery methods, which has been around since early 1990s.
>
> We would like to refer the reviewer again to the sections **Positioning of the paper**  and **Novelty of the ICFL algorithm** in the response to all reviewers. While the algorithm itself is indeed a variation of the orthogonal matching pursuit algorithm, the novelty lies in the usage of this algorithm to extract features from large scale foundation models, and especially in the context of biological image data.
>
> **Comparison with CellProfiler features:**
> >The reason is that ICFL lags behind on almost all metrics behind the original embedding baseline and CellProfiler-derived baselines - As a reader/possible user of the algorithm, I find it less convincing to use ICFL, if it is always performing worse than simpler and older approaches.
>
> We would like to put this comment into perspective as we believe that the results in Figure 3 are a very strong motivation for our approach. CellProfiler features have been developed by domain experts, and thus are building upon decades of biological research. They are (relatively) interpretable, but they are significantly outperformed by black-box deep learning approaches on recall of biological relationships [Kraus et al 2024, others]. ICFL extracts features in an unsupervised manner (with the only signal being used are labels for unperturbed control cells) from a self-supervised trained foundation model. Because these features are extracted in an unsupervised fashion,  they can be used complementary to CellPaint features.
>
> ### Questions:
> >In line 319, why use center crop, instead of the whole image?
>
> The MAE requires us to divide the entire image into crops of specified size, divisible by the token size of 8x8 pixels. There is no particular reason why one could not use the tokens aggregated over all crops per image instead of the center crop only.

---

> > ### Author Response · Authors · 2024-11-24
> > **Follow-up**
> >
> > Dear Reviewer o17p,
> >
> > We hope our rebuttal has addressed your concerns and answered your questions. As the end of the discussion period approaches, we would like to ask if you have any additional questions or concerns, particularly regarding our rebuttal.
> >
> > Thank you once again for your time and thoughtful feedback!

---

> > > ### Author Response · Authors · 2024-11-28
> > > **Response to Reviewer o17p**
> > >
> > > We wanted to follow up regarding the points we addressed in our rebuttal and ensure that your initial concerns and questions have been thoroughly addressed. If there are specific aspects of our rebuttal that you feel require further elaboration or if new questions have arisen, we would be more than happy to address them. Thank you once again for your thoughtful feedback and for contributing to the improvement of our work.
> > >
> > > As we remain committed to addressing your concerns, are there any additional steps we could take to better address your concerns so that you would feel comfortable increasing your score?

---

### Official Review · Reviewer_1UZd · 2024-11-03

**Soundness:** 3
**Presentation:** 3
**Contribution:** 3
**Rating:** 5
**Confidence:** 5

**Summary:**

The paper presents a study of interpretability using sparse dictionary learning for large foundation models of microscopy images. An algorithm is proposed for iterative codebook feature learning (ICFL), which is used to learn a sparse feature space where individual features are presumably more interpretable. Similar methodologies have been used for exploring interpretation of language models, and the presented methodology is an extension for cellular images. The paper presents a quantitative evaluation of how certain known biological and experimental aspects of the images can be identified with the sparse representation, and them compares against manually engineered features. Finally, the paper presents a qualitative evaluation of what the meaning of certain features may be by selecting some which activate with genetic perturbations and channel specific activity.

**Strengths:**

* The presentation of a novel algorithm for learning sparse encodings of token representations. The algorithm could be used for other models and data modalities.
* A quantitative and qualitative evaluation of performance.
* A unique analysis for the specialized domain of cellular images and genetic perturbations.

**Weaknesses:**

1) Are the sparse features really interpretable? Sparse autoencoders emerged as a minimal transformation strategy for interpreting the activations of transformers, which are not directly associated with any semantic meaning. However, this paper does not present strong evidence that the sparse features can be more interpretable than the regular transformer activations for this application domain. There are two reasons for this A) the approach presented in the paper depends on additional token transformations, which makes the reader believe that the interpretation is not really associated to what the model encodes. B) The interpretability scores presented in the quantitative evaluation are based only on a few known labels. Are those labels all that can be interpreted in the models?
2) Aggregated token representations and weak supervision. There are several transformations applied to the tokens before dictionary learning, which obscures the claims made in the paper about interpreting the inner workings of the models. The fundamental question for interpretability is what is the model doing and how we can use that information to keep it under control? Discovering biases and understanding how they are encoded in the model is at the heart of these approaches, but in this paper, the model outputs are first aggregated and decorrelated with PCA whitening using labeled data to proceed with dictionary learning. This contradiction between the claims / motivation of the paper and the actual procedure followed for interpretation is confusing. This also suggests that the interpretation is performed at the level of one downstream task rather than at the level of model activations and inner workings.
3) Monsemanticity? The paper also claims that their approach finds directions that are monosemantic, but the results do not support such claims. The quantitative evaluation is focused on making sure that the learned sparse features capture as much signal as the original features. Monosemantic features would be able to classify these concepts with a single dimension, which is not what the evaluation presents. The results in Fig. 3 seem to indicate that there is still considerable entanglement between features.
4) Not reproducible research. As far as this reviewer can tell, the models and data to reproduce this research are not publicly available. The machine learning community strives for openness and reproducibility, exemplified by the open source, open review, and open access practices. It is a significant downside of this manuscript that the presented results cannot be independently verified.

**Questions:**

* What does it mean to have interpretable features? The paper makes the conclusion that the original token activations are not interpretable, and therefore additional feature transformations are needed. Is this an effect of the lack of a more generic interpretability score for this domain? What is the original model capturing that prevents interpretation without interventions?
* The qualitative evaluation depends on manual interpretation by experts. What would be a more scalable way to make interpretations of morphological variations of cells without having to resort to human annotations?
* To address point 1) and 2) above, how can the authors demonstrate that original features are not as interpretable as the sparse features? This is particularly important before and after the aggregations and transformations used to learn the sparse features. Is it the case that aggregated features after PCA are as interpretable as the sparse codes? What are we gaining in interpretability if these transformations disentangle their meaning from other variation? Please compare feature specificity and other metrics of the original and transformed token features to understand the utility of the proposed algorithm.
* To address point 1) above, what other information can be used to interpret features beyond classification tasks? Something more scalable that requires less manual intervention?
* To address point 3) above, are there any individual features that classify any of the concepts more accurately than random? What percent of the concepts and what percent of the sparse features can be connected in this way? This would clarify if the learned codes are really monosemantic or not. A comparison with original and transformed tokens would be necessary too.
* To address point 4) above, how can others reproduce the results of this research and use them as applications in future work? This research could be conducted entirely with publicly available data by training a community model, for instance. If the research reported in this paper cannot be verified, perhaps this is not scientific progress, but rather a commercial application, in which case other venues are more appropriate for dissemination than ICLR.

---

> ### Author Response · Authors · 2024-11-20
> **Response to Reviewer 1UZd**
>
> **Nature of the main contributions of the paper:**
> We thank the reviewer for raising their critical comments on this paper. We begin by addressing the following comment made by the reviewer.
>
> >  There are several transformations applied to the tokens before dictionary learning, which obscures the claims made in the paper about interpreting the inner workings of the models. The fundamental question for interpretability is what is the model doing and how we can use that information to keep it under control? Discovering biases and understanding how they are encoded in the model is at the heart of these approaches, but in this paper, the model outputs are first aggregated and decorrelated with PCA whitening using labeled data to proceed with dictionary learning.
> > This contradiction between the claims / motivation of the paper and the actual procedure followed for interpretation is confusing. This also suggests that the interpretation is performed at the level of one downstream task rather than at the level of model activations and inner workings.
>
>
> We believe that the reviewer may have a diverging perception of what the main contributions of this paper are, and we would like to revise the manuscript to address the sources of the misunderstandings. As the reviewer correctly points out, the goal of mechanistic interpretability is to understand “what is the model doing and how we can use that information to keep it under control”. However, we would like to emphasize that this is not the main goal of this paper. Instead, we investigate with this paper how the tools from mechanistic interpretability can help us to better understand multi-cell image data. In particular, the goal of this paper is not to find hidden biases in the model, but instead, given the premise that the model lays out abstract concepts as linear directions, to extract features using DL capturing biologically meaningful patterns.  We would like to refer the reviewer also to the section **Positioning of the paper** in the common response addressing all reviewers.
>
> To address the following comments made by the reviewer:
> > This contradiction between the claims / motivation of the paper and the actual procedure followed for interpretation is confusing
> > , which obscures the claims made in the paper about interpreting the inner workings of the models.
>
> May we ask the reviewer to point us to the section in the paper where they got the impression that this paper makes these claims? We would like to address the writing accordingly. Based on the reviewer's comment, we believe that the use of the term ‘interpretable features’ in the caption of Appendix B could be misleading and we will rename this section to Examples of evident patterns captured by the features.
>
>
> **Additional transformations on top of features from MAE:**
>
> > There are two reasons for this A) the approach presented in the paper depends on additional token transformations, which makes the reader believe that the interpretation is not really associated to what the model encodes.
> > There are several transformations applied to the tokens before dictionary learning, which obscures the claims made in the paper about interpreting the inner workings of the models
>
> We thank the reviewer for raising this important concern. We would like provide multiple arguments for why we do not believe that the transformations, i.e. PCA whitening and aggregation of the tokens, stand in contradiction with our analysis.
>
> Finding linear concept directions capturing biologically meaningful information: The underlying assumption that we exploit in this paper is that the model represents abstract concepts as linear directions, which is also the starting point of recent works using DL in mechanistic interpretability. Since we only apply linear transformation before applying DL, the underlying assumption exploited by DL remains unchanged. Moreover, since the goal is primarily to find features capturing biologically striking patterns, we do not see any contradiction in applying these linear transformations, especially as we demonstrate that they improve the scores.

---

> ### Author Response · Authors · 2024-11-20
> **Response to Reviewer 1UZd - Part 2**
>
> Nevertheless, we would like to argue against the claim made by the reviewer that the transformations make it more difficult to deduce insights about the inner working mechanisms of the underlying model in comparison with the pre-processing applied in e.g., [1,2]:
>
> *Token aggregation:* To comment first on token aggregation. Assuming that concepts are laid out by the model as linear directions, these linear directions persist when aggregating the tokens. However, the advantage of aggregating the tokens is that concepts that depend on capturing highly local information, such as whether patch i contains a cell or not, will shrink due to averaging effects. Thus, aggregating the tokens can be seen as a way to reduce the signal to noise ratio that is particularly troublesome in multi-cell image data where batch effects are substantial. Yet, DL should still be able to recover directions of concepts that appear across the entire image, such as exactly the ones we are interested in studying. We are curious to hear the response of the reviewer to our explanation.
>
> *PCA whitening:* While we would agree with the reviewer that the assumptions behind DL, formalized in the superposition hypothesis, is strong, we disagree with the reviewer that applying our weakly supervised PCA transform necessarily results in features that reveal less information about what the model is actually doing compared to ones obtained from apply DL directly. Indeed, in this paper, we propose to minimize the L2 norm after a PCA transform, which can equivalently be understood by applying DL on the original features but minimizing a weighted L2-norm. If we understand correctly, the reviewer now claims that these features are less informative of what the model does compared to the ones obtained when minimizing the l2-norm. However, what justifies the l2-norm as the right norm to minimize? It is very plausible that some concepts are represented by linear directions with smaller magnitudes and others with directions with larger magnitudes. In this case, a data driven approach using weak supervision as we do in this paper is crucial to recover all features. While we do not want to claim that this is indeed the case, we want to challenge the statement by the reviewer that minimizing simply the L2-Norm yields features that are more descriptive of the features actually used by the model. We are curious to hear the thoughts of the reviewer to this response.
>
>
> **Monosemanticity:**
>
> > The paper also claims that their approach finds directions that are monosemantic, but the results do not support such claims. The quantitative evaluation is focused on making sure that the learned sparse features capture as much signal as the original features. Monosemantic features would be able to classify these concepts with a single dimension, which is not what the evaluation presents.
>
> The reviewer highlights an important point, namely that claiming monosemanticity is indeed a strong statement. However,  we usually use monosemanticity in quotation marks and specify what we mean with monosemanticity, such as in Line 537:
> > In our experiments, we found that both ICFL and PCA significantly improve the selectivity or “monosemanticity” of extracted features, compared to TopK sparse autoencoders.
>
> In particular, we do not claim that we find monosemantic features, but instead argue that ICFL together with PCA whitening improves the “monosemanticity” of the features, in the sense that it improves the selectivity score which is indeed a proxy for monosemanticity. Could it be that the reviewer views monosenacity as an attribute that either is present or not, while we treat monosemanticity as an attribute that can also occur only to some extent.
>
> After consideration,  we understand that our use of the term monosemanticity can be misleading, and we will adjust the writing accordingly. In particular,  we will remove the term monosemanticity in the abstract, and thus only write: *“improve the selectivity score of extracted features compared to TopK sparse autoencoders.”*. Nevertheless, we want to clearly object to the reviewers statement that this paper “ also claims that their approach finds directions that are monosemantic”. We explicitly always say that our method improves the ``monosemanticy’’ of the features  and specify  that we mean the selectivity score of the features.

---

> ### Author Response · Authors · 2024-11-20
> **Response to Reviewer 1UZd - Part 3**
>
> **Available labels:**
>
> > B) The interpretability scores presented in the quantitative evaluation are based only on a few known labels. Are those labels all that can be interpreted in the models?
>
> This is indeed an important question. The 5 types of labels that we use in this paper are natural labels of greater significance in the community. In general, one wants to avoid relying too much on labels that require domain expertise for training the probes as generating these labels is expensive. An alternative would be to rely on CellProfiler, which provides a very extensive list of hand-crafted features by domain experts. We use CellProfiler Features in Figure 3. However, limitation in the usage of CellProfiler features is that these features capture visual patterns and not more abstract concepts. One hope of this line of research is that future efforts allow us to  extract features that achieve higher selectivity scores for genetic perturbations than CellProfiler, and thus are individually more predictive of these perturbations.
>
> There are two important points to note in addition to this analysis. First, doing this type of analysis requires the human annotators to precisely know and understand the detailed function of the gene which is knocked out, which means we are naturally restricted for analysis of only a few dozens of genes the function (and morphological consequences) of which we understand and can interpret. Second, observing the images with 20-50 cells each, studying each cell in detail and giving a human verdict about what's going on with that particular cell instance is extremely laborious and doesn't scale well to the repertoires of publicly available cell microscopy datasets, including the ones used in this study. We currently didn't possess the resources of several scientist annotators available, so had to restrict ourselves to a smaller yet more reliable analysis, which is now fully backed up by the quantitative aspect as well.
>
>
> **Entanglement is still present:**
>
> >The results in Fig. 3 seem to indicate that there is still considerable entanglement between features.
>
> We agree with the reviewer. However, to put this comment into perspective, we view it already as a big step that we are able to obtain features that have a selectivity score close to the one of CellProfile features and hope that future works will further improve the selectivity scores. We tried to be transparent about this limitation in the conclusions by writing:
>
>
> > That said, these sparse features are clearly incomplete:we see significant drops in their linear-probing performance on tasks that involve more subtle changes in morphology. It is not clear to what extent this is a limitation of our current dictionary learning techniques, the scale of our models, or whether these more subtle changes are simply not represented linearly in embedding space. Nonetheless, it is clear that the choice of dictionary learning algorithm matters to extract meaningful features.
>
> **Use of proprietary models:**
>
>
> > Not reproducible research. As far as this reviewer can tell, the models and data to reproduce this research are not publicly available. The machine learning community strives for openness and reproducibility, exemplified by the open source, open review, and open access practices. It is a significant downside of this manuscript that the presented results cannot be independently verified.
>
>
> We understand the reviewer's concerns regarding the use of proprietary models, especially given the community's emphasis on reproducibility and openness. However, we believe it's important to note that the field of mechanistic interpretability often relies heavily on proprietary models. Many prominent works in this area, for instance [1,2], use models and frameworks that are not fully open. In general, it is particularly common for large-scale foundational models to use proprietary non-public models. Limiting research in this space based on accessibility to proprietary tools may hinder overall progress and innovation in the field.
>
> Nevertheless, we strongly believe in open science and would also like to contribute to this important community effort. To do so, we’ve made a deliberate effort to use publicly available datasets like RxRx1, RxRx3, and benchmark against the publicly available CellProfiler. Although this does not allow for the full reproducibility of our results directly, our approach allows for comparative benchmarking. By evaluating selectivity scores on RxRx1 and comparing them with CellProfiler features, we’ve created a novel and accessible benchmark that can be independently referenced and utilized. This compromise, we believe, aligns with the spirit of openness while also pushing forward impactful research.

---

> > ### Author Response · Authors · 2024-11-20
> > **Response to Reviewer 1UZd - Part 4**
> >
> > ### Questions:
> >
> > >What does it mean to have interpretable features? The paper makes the conclusion that the original token activations are not interpretable, and therefore additional feature transformations are needed. Is this an effect of the lack of a more generic interpretability score for this domain? What is the original model capturing that prevents interpretation without interventions?
> >
> >
> > May we ask whether the response in the paragraph above Nature of the main contributions of the paper answers their first question. To respond to the second question, may we ask the reviewer where we made the claim in the paper that the original features are not interpretable? We simply observe that PCA whitening improves the scores in Figures 2-3 and we therefore motivate its use. These are standard scores also used e.g., in the analysis in [1]. To answer the last question, may we ask the reviewer to be more precise in what they mean by “without interventions”?
> >
> >
> >
> >
> > > The qualitative evaluation depends on manual interpretation by experts. What would be a more scalable way to make interpretations of morphological variations of cells without having to resort to human annotations?
> > >To address point 1) above, what other information can be used to interpret features beyond classification tasks? Something more scalable that requires less manual intervention?
> >
> > We thank the reviewer for fair raising this very important question. To some extent we have to leave a precise answer as an important task for future work. However, we nevertheless would like to sketch a non-exhaustive list of possible directions. On the one hand, the scores used in this paper like the sensitivity score provide some signal that can help for a first selection. Furthermore, more advanced statistical approaches can be developed, such as studying the dependencies of genes that strongly activate a specific feature and compare them with known gene-gene interactions from publicly available databases. This direction is particularly promising as it might help to determine which potentially unknown gene-gene interactions deserve additional investigations by researchers. On the other hand, the use of multi-modal LLMs is a promising direction that could assist researchers in a first pre-selection.
> >
> > > To address point 1) and 2) above, how can the authors demonstrate that original features are not as interpretable as the sparse features? This is particularly important before and after the aggregations and transformations used to learn the sparse features. Is it the case that aggregated features after PCA are as interpretable as the sparse codes? What are we gaining in interpretability if these transformations disentangle their meaning from other variation? Please compare feature specificity and other metrics of the original and transformed token features to understand the utility of the proposed algorithm.
> >
> > We would like to give a detailed answer to your question. However, provided the response in the previous paragraphs, may we ask the reviewer to be more precise about their concerns? In particular, what do they mean by “demonstrate that original features are not as interpretable as the sparse features”. To address this comment: "Please compare feature specificity and other metrics of the original and transformed token features to understand the utility of the proposed algorithm.” - We provided multiple scores such as probing and the selectivity score, that allow us to evaluate whether these features capture biologically meaningful information.
> >
> > > To address point 3) above, are there any individual features that classify any of the concepts more accurately than random? What percent of the concepts and what percent of the sparse features can be connected in this way? This would clarify if the learned codes are really monosemantic or not. A comparison with original and transformed tokens would be necessary too.
> > To address the first question.
> >
> > The reviewer raises an important question. We report the number of classes for each task but not the accuracy of a random classifier. RxRx1 has over 1k features that are almost balanced, and thus any random classifier has an accuracy of about 1/1000. In contrast, we find features having selectivity scores for some labels above 0.4 (average selectivity score in Figure 2), which implies that for those pairs of features and labels we obtain a balanced binary classification accuracy (one against all) of more than 40%.
> >
> >
> > [1] Gao, Leo, et al. "Scaling and evaluating sparse autoencoders." arXiv preprint arXiv:2406.04093 (2024).
> >
> > [2] Bricken, Trenton, et al. "Towards monosemanticity: Decomposing language models with dictionary learning." Transformer Circuits Thread 2 (2023).

---

> > > ### Author Response · Authors · 2024-11-24
> > > **Follow-up**
> > >
> > > Dear Reviewer 1UZd,
> > >
> > > We hope our rebuttal has addressed your concerns and answered your questions. As the end of the discussion period approaches, we would like to ask if you have any additional questions or concerns, particularly regarding our rebuttal.
> > >
> > > Thank you once again for your time and thoughtful feedback!

---

> > ### Comment · Reviewer_1UZd · 2024-11-27
> >
> > Thanks again for clarifying the token aggregation and validity of the PCA whitening. I encourage the authors to help the readers understand these choices as important problems in this domain and clarify the motivation and significance of them.
> >
> > Regarding the “monosemanticity” term, my confusion comes from the definition of the term itself. It’s supposed to be one (mono) meaning (semantic). If the feature is not monosemantic, I understand it as polysemantic or entangled with other meanings. Monosemanticity cannot be understood as a continuum, as the authors suggest. Also, improving monosemanticity means that more features become monosemantic, not that one features turns a little more monosemantic. In previous literature, this is evaluated with density and conditional probability functions that convincingly show how one sparse code almost exclusively identifies a topic. Of course, in NLP tokens and topics are discrete while in images that’s not the case. But this is another example where terms borrowed from NLP may be against the readability of this paper.

---

> ### Comment · Reviewer_1UZd · 2024-11-27
>
> Thank you for addressing my comments and for clarifying. I acknowledge that the common response to all reviewers was helpful, including the position of the paper and scientific relevance. In addition, I see value in the proposed algorithm for general purpose mechanistic interpretation.
>
> To clarify my comment about contradiction between claims / motivation, I believe the paper reuses terminology from mechanistic interpretability in natural language processing which results in mismatched expectation and results. For instance, the fact that the authors use the word “token” for the superposition hypothesis and subsequent formulations, to later on find out that the paper is not doing a single token analysis, is confusing. The use of the “monosemanticity” term (more below) was also confusing. This is a pattern where the paper seems to reuse ideas from NLP without critically thinking how that matches the problem at hand. I think there is a unique opportunity to highlight the challenges in this domain, inspired by success in NLP, but without suggesting that the same thing is happening.
>
> I appreciate the clarification about PCA being a linear transformation too and the detailed discussion about optimizing L2 vs a weighted L2. I agree with the authors that this is very unique and it is an important fundamental question to address. However, this is not clearly motivated in the paper beyond “they improve the scores”. This could be an example where differences in domain may result in rich opportunities to advance the field as mentioned above.

---

> ### Comment · Reviewer_1UZd · 2024-11-27
>
> To clarify the comment about “without interventions”, early experiments in mechanistic interpretability attempted to understand if the original activations of the model are interpretable or not. Later on, sparse coding was introduced to identify feature combinations that were more “monosemantic”. This work makes the assumption that sparse codes are necessary, but the author’s argument could be strengthened if they show that the original feature space is not as interpretable as the codes they learn in this domain.
>
> Finally, it's true that previous work have used and relied on proprietary models to conduct the evaluations, but even the papers that the authors cite have conducted experiments in open models (e.g. GPT2) for the community to reproduce and continue exploring such research directions. This is something that this paper does not have in its current form.
>
> Thank you for offering clarifications about the random classifier and challenges of what is interpretable in this domain. Overall, I appreciate the responses and the work behind this submission. I have modified the rating from 3 to 5 in light of these. Please, update the manuscript with the suggested changes and clarifications.

---

> > ### Author Response · Authors · 2024-11-28
> > **Response to Reviewer 1UZd**
> >
> > We sincerely appreciate your engagement with our work and the time you've taken to carefully review and reconsider our submission. We are actively working on incorporating the suggested changes and clarifications, alongside implementing ongoing feedback from other reviewers, all of which will be reflected in the final manuscript.
> >
> > We noticed the change in your score from 6 to 5 and wanted to kindly reach out for clarification. If there are specific remaining concerns or additional points of confusion, we would be more than happy to provide further explanations or adjustments to address them.
> >
> > Your detailed feedback has already helped us significantly improve the clarity and positioning of the paper, and we are committed to ensuring that the contributions are presented in a manner that fully aligns with your expectations. If the current score reflects unresolved questions or uncertainties, we’d greatly value the opportunity to address them directly.
> >
> > Building on the constructive engagement so far, are there any additional steps we could take to better address your concerns so that you would feel comfortable increasing your score?

---

### Official Review · Reviewer_SfZF · 2024-11-03

**Soundness:** 3
**Presentation:** 3
**Contribution:** 4
**Rating:** 8
**Confidence:** 4

**Summary:**

The authors propose a Dictionary Learning method to discover unknown concepts from Foundation Models that have been pre-trained on image-based phenotypic screening data.  Specifically, the goal is to learn a new, sparse feature representation, such that individual features align with interpretable biological concepts.  To this end, they propose a new method called Iterative Codebook Feature Learning (ICFL) and benchmark it against a competitive approach: TopK Sparse Autoencoders (Topk SAE).  Their proposed method is closely related to orthogonal matching pursuit algorithm, and they demonstrate that it reduces the number of dead/unused features relative to TopK SAE on their dataset.

Using this method, the authors demonstrate that sparse dictionaries can extract biologically meaningful concepts such as cell type and functional genetic perturbations, and that ICFL features demonstrate higher selectivity for these concepts than TopK SAE.  They also demonstrate that PCA whitening has a significant impact on the selectivity of the learned features.  In addition, the paper shows qualitative examples of images that correspond to specific concepts, and provides possible interpretations by way of heat maps highlighting local correlation between the token map and feature direction.

**Strengths:**

The authors address a challenging and important problem in drug discovery, namely the lack of interpretability of learned features from large-scale phenotypic screens.  This is a particularly challenging problem, because it involves interpreting "black box" deep learning models in a domain where there are not clear examples of known concepts to draw upon.  By contrast, in natural images, objects such as cars and animals provide numerous intuitive visual concepts.  To address this challenge, the authors leverage 5 classification tasks with qualitatively different labels: (1) cell type, (2) experimental batch, and (3-5) genetic perturbations.  By evaluating the performance (eg, via linear probing) of their learned features on these tasks, they can ascribe some biological significance to the learned concepts.  Of particular relevance are features that align with the functional genetic perturbation labels.

The paper then proposes a novel method called Iterative Codebook Feature Learning (ICFL) to improve upon existing Dictionary Learning methods which often learn imbalanced features, where some features are unused (so called "dead latents"), and other features are over-used.  Based on the 5 tasks listed above, the paper evaluates the performance of ICFL and TopK SAE on linear probing.  In addition, they evaluate the selectivity of features learned from these two methods, demonstrating that ICFL both retains more information and has more selective features than TopK SAE.  They also observe that PCA whitening has a significant effect on the quality of learned features.

Based on the above framework, the authors provide an analysis of the number of features that have selectivity above a threshold (0.1, 0.2 and 0.5) for their 5 different task labels.  Interestingly, while a large fraction of features have high selectivity for cell type (23 labels), very few features have selectivity for functional gene group (39 labels).  This highlights the remaining challenge of learning concepts that are specific to subtle genetic perturbations.

Overall, the application of dictionary learning to image-based phenotypic screening is a promising direction, and the paper includes multiple insightful analyses.  While there remains significant progress to be made towards extracting meaningful concepts from images of cells under genetic perturbation, the paper represents a significant and original contribution to this topic.

**Weaknesses:**

The paper proposes a new method for Dictionary Learning, called Iterative Codebook Feature Learning, specifically to address the challenge of inadvertently learning "dead latents" by existing methods.  Their method appears to very effectively achieve this goal (Table 1); however, this contribution would be more significant if the authors included an evaluation on additional datasets, such as those used in Gao et al.

In Figure 1, the authors provide multiple examples of images ranked by their alignment with different learned concepts.  However, it's difficult to conclude how distinct the learned concepts are.  For example, Features A, B and C (and potentially numerous other features) could all be influenced by the total cell count in the images.  In addition, Figure 2 d-f suggest that numerous different features have a similar correlation with labels such as Cell Type, Perturbation ID, and Functional Gene Group.

The authors provide a qualitative analysis of their method's ability to identify novel biological concepts in Figures 4 and 5.  For example, in Figure 4, they investigate a feature that is strongly correlated with gene knock outs from the adherens junctions pathway.  By visualizing  heat map of the inner product of individual tokens with the learned feature, they conclude that specific cells within each image have escaped the perturbation, exhibiting normal phenotypes, and correspondingly are not highlighted in the heat maps.  However, this qualitative analysis could inadvertently suffer from confirmation bias.  I suggest a more rigorous analysis, such as having a blinded expert identify cells that are not exhibiting an expected phenotype for a genetic perturbation.  This would produce a cell-level annotation of outlier cells, amongst other cells that are exhibiting the intended phenotype.  Based on this, it would be interesting to evaluate the correspondence of token-level heat maps with this annotation.


The clarity of Figure 2 could be improved.  Here are some suggestions:

1) In Fig 2a, tasks are listed by number ((1)-(5)); however this isn't clearly described in the figure caption.  I suggest either naming the tasks explicitly in Fig 2a, or including a reference in the figure caption.

2) There appears to be a typo in the legend for Fig 2a-c.  The dashed line is listed as "w/o pcaw" and there is a missing line for "w pca".


Additional minor edits:

1) Section 6.2: "...image taken form a subset of the public..." -> "...image taken from a subset of the public..."

**Questions:**

I was confused by the histograms in Figure 3 a-d.  In particular, for some genetic relationships (a,c) the result of paired genetic perturbations was to reduce the mean cosine similarity, yielding a negative value.  Is this expected for these specific gene pairs?  It would be helpful to include additional discussion on the expected cosine-similarity for each pairs of perturbations.

---

> ### Author Response · Authors · 2024-11-20
> **Response to Reviewer SfZF**
>
> **Benchmarking of ICFL:**
> > Their method appears to very effectively achieve this goal (Table 1); however, this contribution would be more significant if the authors included an evaluation on additional datasets, such as those used in Gao et al.
>
> We agree with the reviewer, but view such an analysis outside of the scope of this paper. We would like to refer the reviewer to the section **Benchmark comparison of ICFL** in the comment addressing all reviewers.
>
> **Confirmation bias of qualitative analysis:**
> > By visualizing a heat map of the inner product of individual tokens with the learned feature, they conclude that specific cells within each image have escaped the perturbation, exhibiting normal phenotypes, and correspondingly are not highlighted in the heat maps. However, this qualitative analysis could inadvertently suffer from confirmation bias.
>
> While we generally agree that there are serious risks of confirmation bias with any qualitative output of an interpretability method so it is good to be wary, we disagree with this characterization. The fact that some of the cells escaped the perturbation is evident from the images themselves, and the fact that the inner product is significantly smaller for these cells is evident from the heatmap. With the qualitative analysis we demonstrate that we can find features that capture highly non-random patterns, similar to the ‘’golden gate bridge’’ feature extensively discussed in [1].
>
> **Extensions of the qualitative analysis:**
> > I suggest a more rigorous analysis, such as having a blinded expert identify cells that are not exhibiting an expected phenotype for a genetic perturbation. This would produce a cell-level annotation of outlier cells, amongst other cells that are exhibiting the intended phenotype. Based on this, it would be interesting to evaluate the correspondence of token-level heat maps with this annotation.
>
> We thank the reviewer for sharing this advanced and intriguing approach for evaluating the features. We took a major step towards addressing your concern in section **Interpretability analysis** in the response addressing all reviewers. Furthermore, we strongly believe that developing robust processes for assessing the biological value of features extracted from foundation models along the lines of what the reviewer described, and especially ones that involve domain experts, is a very valuable and important contribution on its own.
>
> >The clarity of Figure 2 could be improved. Here are some suggestions:
>
> We thank the reviewer for the suggestions and will implement them in a revised version of this paper.
>
>
> ### Questions:
> >I was confused by the histograms in Figure 3 a-d. In particular, for some genetic relationships (a,c) the result of paired genetic perturbations was to reduce the mean cosine similarity, yielding a negative value. Is this expected for these specific gene pairs? It would be helpful to include additional discussion on the expected cosine-similarity for each pairs of perturbations.
>
> We plot the correlation of tokens from specific genetic perturbations with selected feature directions, showing that the feature directions can separate the distribution from the distribution of all tokens. We would like to stress that the features learned by the ICFL algorithm can be both positive and negative, and the algorithm has no specific preference towards positive values (in strong contrast to TopK SAE). Indeed, one can simply replace the feature directions of the ICFL algorithm w_i with -w_i  (Section 4) with the only impact that z_i becomes -z_i, however, without changing the reconstruction \hat x_i. May we ask the Reviewer whether this explanation answers their question?

---

> > ### Author Response · Authors · 2024-11-24
> > **Follow-up**
> >
> > Dear Reviewer SfZF,
> >
> > We hope our rebuttal has addressed your concerns and answered your questions. As the end of the discussion period approaches, we would like to ask if you have any additional questions or concerns, particularly regarding our rebuttal.
> >
> > Thank you once again for your time and thoughtful feedback!

---

> > > ### Comment · Reviewer_SfZF · 2024-11-27
> > >
> > > > We plot the correlation of tokens from specific genetic perturbations with selected feature directions, showing that the feature directions can separate the distribution from the distribution of all tokens. We would like to stress that the features learned by the ICFL algorithm can be both positive and negative, and the algorithm has no specific preference towards positive values (in strong contrast to TopK SAE). Indeed, one can simply replace the feature directions of the ICFL algorithm w_i with -w_i (Section 4) with the only impact that z_i becomes -z_i, however, without changing the reconstruction \hat x_i. May we ask the Reviewer whether this explanation answers their question?
> > >
> > > Yes, thank you for providing clarification, and for including the more rigorous Interpretability Analysis in the supplemental.
> > >
> > > I don't have any further questions.

---

### Author Response · Authors · 2024-11-20
**General response to all reviewers**

We sincerely thank the reviewers for their detailed reviews. We also appreciate the constructive feedback aimed at refining our manuscript. The following comments address prominent issues highlighted by the reviewers. We will address specific feedback/comments in our individual responses.



## Positioning of the paper
Since this is an interdisciplinary work bringing together concepts from different domains, we would like to emphasize again the main contribution and positioning of this paper: we explore how **tools from mechanistic interpretability** can be used in the **context of scientific discovery**. In contrast to works from mechanistic interpretability, our primary aim is to explore whether these tools help us to find features capturing biologically relevant and potentially undiscovered concepts. This aim is reflected in the reviews of Reviewers SfZF and 2RjZ:

> Specifically, the goal is to learn a new, sparse feature representation, such that individual features align with interpretable biological concepts.
> Is the desired goal that by looking through the learned features, a researcher might find a single concept that captures something complex and unexpected, warranting further study?

And, as we write in the introduction,

> given that models can detect subtle differences in images (even those that are very challenging for human experts to interpret), we might hope that we can use these techniques to better understand subtle differences.

More precisely, we explore the ability of dictionary learning type of techniques to extract features from large-scale foundation models that are correlated with biological signals like cell type or genetic perturbation. In particular, Figure 3 shows that our features achieve a selectivity score close to that of hand-crafted features (by domain experts).


## Scientific relevance
While the scientific relevance was clear to some reviewers, e.g, Reviewer SfZF:

> Overall, the application of dictionary learning to image-based phenotypic screening is a promising direction, and the paper includes multiple insightful analyses. While there remains significant progress to be made towards extracting meaningful concepts from images of cells under genetic perturbation, the paper represents a significant and original contribution to this topic.

Reviewers 2RjZ and eDB6 raised questions about the scientific relevance and concrete applications:

> More generally, how do you see the results of this method being used in biological research?
> Overall the contribution seems novel and non-trivial, however, its future scientific significance appears to be vague to me at this point.

We thank the reviewers for raising this important point. To answer Reviewer 2RjZ’s question: *“Is the desired goal that by looking through the learned features, a researcher might find a single concept that captures something complex and unexpected, warranting further study?”* -  Yes, precisely, this is one of the concrete applications. Given that we know that unsupervised representation learning approaches significantly outperform CellProfiler (hand crafted features) for recall of biological relationships [1], and that transformer representations often encode concepts linearly; we aim for a technique that enables scientists to describe genetic perturbations in terms of learned concepts rather than hand-crafted features. Novel techniques in this direction could have an actual impact on how scientists study genetic (and potentially extend to compound) perturbations.

---

> ### Author Response · Authors · 2024-11-20
> **General response to all reviewers - Part 2**
>
> ## Novelty of the ICFL algorithm
> To extract features from our foundation model, we use TopK-SAE, which is a SOTA technique proposed in the mechanistic interpretability community. Furthermore, we find that our own algorithm, ICFL, which is indeed just a variant of the orthogonal matching pursuit yields substantially better features. Since we believe that this finding may be of independent interest for the mechanistic interpretability community, we mention it as a separate technical contribution. Different reviewers raised concerns about the novelty of the ICFL algorithm:
>
> > Technical novelty At the end of the day, ICFL is a variation of Orthogonal Matching Pursuit and also restricted to L0 recovery methods, which has been around since early 1990s.
> > With respect to machine learning methodology, the technical novelty of the paper is somewhat limited. The overall problem and approach follows TopK, except rather than optimizing the specific TopK formulation, an orthogonal matching pursuit approach is used instead.
>
> We would like to put these comments into perspective and stress that most well-known results in mechanistic interpretability use TopK-SAE [2] or L1-SAE [2] which were both already well known [4] algorithms. However, while these methods are simple to implement in deep learning frameworks, it is not obvious that they lead to good disentanglement results (especially in the context of images where there have been fewer successful applications), and our results show you can get significantly more disentangled transformer representations with matching pursuit. To ensure that this point is clear, we will add 2 lines to the revised version of the paper.
>
>
> ## Benchmark comparison of ICFL
> Reviewers SfZF and 2RjZ also raised concerns about the scope of the benchmark comparison against TopK-SAE:
>
> > Their method appears to very effectively achieve this goal (Table 1); however, this contribution would be more significant if the authors included an evaluation on additional datasets, such as those used in Gao et al.
> >  The authors restrict the analyses on MAE-pretrained ViTs, whereas if the authors also perform similar analyses on other types of SSL-pretrained ViTs for this domain, I think that would strengthen the argument that ICFL is robust.
> > ICFL was shown to be superior to TopK on some quantitative measures, but these measures are proxies and not an end goal in themselves. Qualitative results were only given for ICFL. Was any qualitative analysis done on the TopK learned features?
>
> In this paper, we favor an in-depth analysis of features extracted from one specific model over multiple models. This is common in the existing literature (see e.g., [2,3]) and has two reasons: the first one is the access to such models, the corresponding datasets and the necessary infrastructure to run these experiments. We highlight here that many works e.g., asdf use proprietary models. The second reason is that assessing the quality of the extracted features is a highly domain-specific task. In this paper we focus on multi-cell image data, and propose an evaluation strategy specifically targeted for this data modality.
>
> To further comment on the metrics used in this paper to compare TopK-SAE with ICFL, we use the number of dead features (Table 1), the linear probing scores (Figure 2), the selectivity scores (Figure 2 and 3), and the reconstruction error (Figure 2 and 7). These scores are similar to the ones used in [2] to evaluate SAEs, with the addition of the selectivity score. Regarding a qualitative analysis comparing ICFL with TopK-SAE, we do not think that such an analysis is the right approach for comparing the two algorithms; confirmation bias is risky in any interpretability work, and this is especially true in cell imaging where the differences between images can be extremely subtle and hard to interpret, even for human experts. We would like to stress that the main purpose of the qualitative analysis in Section 7 is to illustrate striking patterns captured by the features and therefore motivate the potential of this line of research as a whole.
>
> [1] Kraus, Oren, et al. "Masked Autoencoders for Microscopy are Scalable Learners of Cellular Biology." Proceedings of the IEEE/CVF Conference on Computer Vision and Pattern Recognition. 2024.
>
> [2] Gao, Leo, et al. "Scaling and evaluating sparse autoencoders." arXiv preprint arXiv:2406.04093 (2024).
>
> [3] Bricken, Trenton, et al. "Towards monosemanticity: Decomposing language models with dictionary learning." Transformer Circuits Thread 2 (2023).
>
> [4] Makhzani, Alireza, and Brendan Frey. "K-sparse autoencoders." arXiv preprint arXiv:1312.5663 (2013).

---

> > ### Author Response · Authors · 2024-11-20
> > **Interpretability analysis: Figures are in the supplementary material**
> >
> > To address some of the reviewers’ questions concerning the qualitative analysis form Section 7, we ran additional experiments that we provide in the **supplementary materials** to enrich the qualitative findings with more robust, quantitative analysis to further support the claims and patterns already shown in the imaging data from Figure 5. We summarize the findings of our extended analysis in the following Table 1.
> >
> > ## Manual labeling effort
> >
> > All 5 images from Figure 5 of the main paper, comprising 121 cells in total, were subjected to scoring by a human expert annotator who categorized the cells into 2 subgroups: single cells reflecting the real perturbation or single cells appearing to have escaped the perturbation, which exhibit a control-like phenotype. Out of 31 cells which were scored as control-like by the human annotator, the token level heatmap areas corresponding to these single cells were “darker” in 27 instances. The darker cell areas correspond to tokens with lower alignment to the general direction of the image, and hence are indicative of lower importance of these areas in the image overall. Each cell was only scored as “dark” or “bright” in the token heatmap by an expert annotator after the manual cell labeling was completed. We report that the token heatmaps are capable to “recall” 87% of the expert annotator labels overall.
> >
> >
> > | Image (as labeled in Figure 5) | **Manual human labeling**                 |                                     |                                   | **SAE token heatmap highlighted cells as “dark”** | **Recall (“dark” cells over all human-labeled controls)** |
> > |--------------------------------|-------------------------------------------|-------------------------------------|-----------------------------------|---------------------------------------------------|----------------------------------------------------------|
> > |                                | **Total cells detected**                  | **Perturbation-reflecting cells**   | **Control-like [escaped] cells**  |                                                   |                                                          |
> > | *A*                            | 28                                        | 22                                  | 6                                 | 6                                                 | 100.0%                                                     |
> > | *B*                            | 34                                        | 26                                  | 8                                 | 6                                                 | 75.0%                                                      |
> > | *C*                            | 20                                        | 17                                  | 3                                 | 3                                                 | 100.0%                                                     |
> > | *D*                            | 20                                        | 13                                  | 7                                 | 6                                                 | 85.7%                                                    |
> > | *E*                            | 19                                        | 12                                  | 7                                 | 6                                                 | 85.7%                                                    |
> > | **Total:**                     | **121 cells**                             | **90 cells**                        | **31 cells**                      | **27 cells**                                      | **87.1%**                                                |

---

> > > ### Author Response · Authors · 2024-11-20
> > > **Interpretability analysis: Figures are in the supplementary material - Part 2**
> > >
> > > ## Response to reviewers comments
> > >
> > > To respond to Reviewer SfZF’s comment:
> > > >  However, this qualitative analysis could inadvertently suffer from confirmation bias. I suggest a more rigorous analysis, such as having a blinded expert identify cells that are not exhibiting an expected phenotype for a genetic perturbation. This would produce a cell-level annotation of outlier cells, amongst other cells that are exhibiting the intended phenotype. Based on this, it would be interesting to evaluate the correspondence of token-level heat maps with this annotation.
> > >
> > > We thank the reviewer for this suggestion. We provide such an extended quantitative analysis for the 5 images from the Figure 5 of the main paper in supplementary materials (Figure 1 and Table 1) as well as in the Table 1 above. We will add this analysis to the revised version of the paper. As we can see, the labels produced by using the features from the SAE strongly align with the labels provided by our domain expert.
> > >
> > >
> > > To respond to Reviewer o17p’s comment
> > > > The paper makes the conclusion that the original token activations are not interpretable, and therefore additional feature transformations are needed.
> > >
> > > We appreciate the reviewer’s suggestion and agree that such a comparison adds significant value to the paper. Accordingly, we have included an analysis in Figure 2 of the supplementary material using the "raw" embeddings from the residual stream. Our findings reveal that these features are unable to effectively predict whether a cell exhibits morphological changes. In contrast, the linear feature direction of the SAE feature analyzed in Figure 1 is highly predictive as also shown in Table 1.

---

### Meta-Review · Area_Chair_xS6j · 2024-12-21

**Metareview:**

The paper proposes a dictionary learning  method for microscopy images with the goal of
extracting concepts. They introduce an algorithm named Iterative Codebook Feature Learning (ICFL).
Their findings suggest that ICFL improve feature
 selectivity compared to  sparse autoencoders.

While the paper shows promising results, reviewers have concerns about
the technical novelty of ICFL, the interpretability of extracted features.  I tend to concur with
these concerns especially the one on the technical novelty given the huge literature
in the signal processing community on sparse recovery and on multiple measurement sparse recovery.
As such, I believe that the paper still needs better literature positioning and  some improvements before reaching an acceptable level.

**Additional Comments On Reviewer Discussion:**

reviewers have acknowledged the rebuttals and issues about  technical novelty have been still pointed out and not cleared by the rebuttals.

---

### Decision · Program_Chairs · 2025-01-22

Reject